# A New Linear Scaling Rule for Differentially Private Hyperparameter Optimization

## Abstract

A major direction in differentially private (DP) machine learning is DP fine-tuning: pretraining a model on a source of public data and transferring the extracted features to downstream tasks. This is an important setting because many industry deployments fine-tune publicly available feature extractors on proprietary data for downstream tasks. In this paper we propose a new linear scaling rule, a hyperparameter optimization algorithm that privately selects hyperparameters to optimize the privacy-utility tradeoff. A key insight into the design of our method is that our new linear scaling rule jointly increases the step size and number of steps as $\varepsilon$ increases. Our work is the first to obtain state-of-the-art performance on a suite of 20 benchmark tasks across computer vision and natural language processing for a wide range of $\varepsilon \in [0.01, 8.0]$ while accounting for the privacy cost of hyperparameter tuning.

## 1 Introduction

Industry deployments make use of pretrained models [79] by fine-tuning on task-specific datasets [35; 6; 69] and serving consumer applications that span the range of modalities from portraiture [65] to chatbots [44]. A crucial component of interfacing machine learning models closely with user data is ensuring that the process remains *private* [74], and Differential Privacy (DP) is the gold standard for quantifying privacy risks and providing provable guarantees against attacks [20]. DP implies that the output of an algorithm e.g., the final weights trained by stochastic gradient descent (SGD) do not change much if a single datapoint in the dataset changes.

**Definition 1.1** (Differential Privacy)**.** A randomized mechanism $\mathcal{M}$ with domain $\mathcal{D}$ and range $\mathcal{R}$ preserves $(\varepsilon, \delta)$-differential privacy iff for any two neighboring datasets $D, D' \in \mathcal{D}$ and for any subset $S \subseteq \mathcal{R}$ we have $\Pr[\mathcal{M}(D) \in S] \leq e^{\varepsilon} \Pr[\mathcal{M}(D') \in S] + \delta$

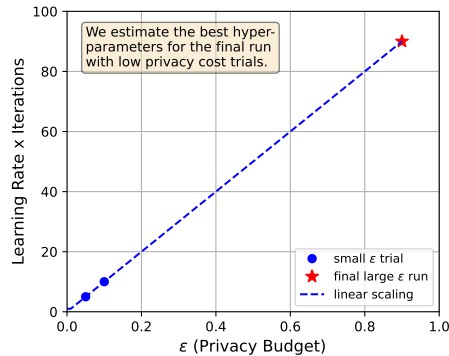

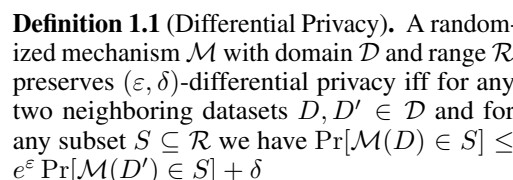

Figure 1: Our new linear scaling rule first does a small number of trials with a very small privacy budget, then does a small number of trials with a slightly larger privacy budget, and finally does linear interpolation through the optimal hyperparameters from these low-cost runs up to the final privacy cost

where $D$ and $D'$ are neighboring datasets if they differ in a single entry, $\varepsilon$ is the privacy budget and $\delta$ is the failure probability.

Differentially Private Stochastic Gradient Descent (DP-SGD) [72; 1] is the standard privacy-preserving training algorithm for training neural networks on private data, with an update rule given by $w^{(t+1)} = w^{(t)} - \frac{\eta_t}{|B_t|} \left( \sum_{i \in B_t} \frac{1}{C} \mathbf{clip}_C(\nabla\ell(x_i, w^{(t)})) + \sigma\xi \right)$ where the changes to SGD are the per-sample gradient clipping $\mathbf{clip}_C(\nabla\ell(x_i, w^{(t)})) = \frac{C \times \nabla\ell(x_i, w^{(t)})}{\max(C, ||\nabla\ell(x_i, w^{(t)})||_2)}$, and addition of noise sampled from a $d$-dimensional Gaussian distribution $\xi \sim \mathcal{N}(0, 1)$ with standard deviation $\sigma$. These steps alter the bias-variance tradeoff of SGD and degrade utility, creating a challenging privacy-utility tradeoff. Recent work has made significant progress in closing the gap in performance between private and non-private fine-tuning of transformer-scale models [46; 52; 7; 51], but a key problem presents a concrete obstacle to implementing DP algorithms to power real-world consumer-facing machine learning applications.

The privacy analysis of current approaches for private training does not account for the cost of hyperparameter tuning, and DP-SGD additionally increases the hyperparameter tuning burden compared to vanilla SGD. These hyperparameters include the learning rate schedule, the clipping bound, the batch size, and the amount of noise to add at each iteration. Because private training introduces additional hyperparameters, biases optimization by clipping the gradient, and imposes privacy-utility tradeoffs for existing hyperparameters, it is challenging to apply hyperparameter selection strategies from non-private training, even on the same dataset. Furthermore prior SOTA work in private training does not use similar hyperparameters as non-private training so hyperparameter search algorithms cannot be leveraged from the broader literature. More specifically, conventional non-private training uses SGD with momentum [61] or AdamW [36] to train for hundreds of epochs. However, training for additional iterations in DP-SGD requires adding additional noise [27], and taking large step sizes (such as with momentum) with low signal-to-noise ratio (SNR) can destabilize training [3]. Prior

Figure 2: We compare the best private and best non-private test accuracy performances of our method to prior work using models pretrained on ImageNet-21k and fine-tuned on CIFAR10 and CIFAR100 datasets. Our results at $\varepsilon = 1$ include the cost of hyperparameter tuning via applying the linear scaling rule at $\varepsilon \in [0.01, 0.1]$.

| Dataset | Approach | $\varepsilon = 1$ | $\varepsilon = \infty$ |
|---------|----------|-------------------|------------------------|
| CIFAR10 | Ours | **99.00** | 99.00 |
| | [51] | 96.30 | 96.60 |
| | [7] | 96.70 | 97.40 |
| | [9] | 95.00 | 96.40 |
| | [15] | 94.80 | 96.60 |
| CIFAR100 | Ours | **89.62** | 91.57 |
| | [51] | 82.70 | 85.29 |
| | [7] | 83.00 | 88.40 |
| | [9] | 73.70 | 82.10 |
| | [15] | 67.40 | 81.80 |

work aims to minimize the amount of noise that is added during training by utilizing early stopping, training for as little as a single iteration [51]. Prior work has either fixed these hyperparameters without explanation [7] or performed an extensive search to find the best values [15], but the hundreds of trials of hyperparameter tuning [51] go unaccounted for in the privacy analysis.

We propose a new linear scaling rule (Alg. 1, Fig. 1) that automatically selects hyperparameters to optimize the privacy-utility tradeoff of private fine-tuning. In particular, as our privacy budget increases from $\varepsilon = 0 \to \infty$, we increase the step size and number of steps. Our method accounts for the privacy cost of hyperparameter selection by allotting a small portion of the budget to find the best hyperparameters at $\varepsilon \ll 1$ and scaling these up to $\varepsilon = 1$. We summarize our contributions:

- We demonstrate that our new linear scaling rule reduces the computation and privacy cost of hyperparameter optimization by an order of magnitude without sacrificing performance

- Linear scaling can obtain new SOTAs for both full fine-tuning and linear probing of both convolutional and transformer architectures across 20 vision and language tasks

- We compare four model architectures for a set of five vision benchmarks and find that the private-non private utility gap decreases as models improve, with the best model across all five tasks obtaining lossless performance of 99% accuracy for $\varepsilon = 1$ on CIFAR10

- We find that linear scaling is robust to domain shifts between private training and test data

- We find that models trained with our method can provide good performance even when there is a large shift between public and private data

- We validate that models trained with our method can perform well for zero-shot classification

- We provide our code as a part of our empirical evaluation.

---

**Algorithm 1** DP-SGD with Linear Scaling

---

**Inputs:** Private dataset $\mathcal{D}$, open source feature extractor F, number of classes $C$, privacy budget $\varepsilon$, momentum $\rho = 0.9$, first search privacy budget $\varepsilon_0$, second search privacy budget $\varepsilon_1$

Perform first hyperparameter search to obtain the best possible value of $r_0$ within the first privacy budget $\varepsilon_0$

Perform second hyperparameter search initialized at $r_1^* = \frac{\varepsilon_1}{\varepsilon_0} \cdot r_0$ to obtain the best possible value of $r_1$ within the second overall privacy budget $\varepsilon_1$

Perform linear interpolation to estimate the slope $\alpha$ and bias $b$ of the line $r = \alpha\varepsilon + b$ given $(r_0, \varepsilon_0), (r_1, \varepsilon_1)$

Set $r^* = \alpha\varepsilon_f + b$ given the estimated linear interpolation

Extract features from $\mathcal{D}$ using F: $\mathcal{X} = \mathrm{F}(\mathcal{D})$

Zero-initialize classifier $w \leftarrow 0_{C \times d}$

Decompose the total step size $r$ given by linear scaling into $r = \eta \times T$

Use privacy loss variable accounting to calibrate noise parameter $\sigma$ given $\varepsilon$

**for** $i = 1, 2, \ldots, T$ **do**

    Compute full-batch gradient according to Eq. 1 $\nabla^{(i)} = \frac{1}{|D|} \left( \sum_{i \in D} \mathbf{clip}_1(\nabla\ell(x_i, w^{(i)})) + \sigma\xi \right)$

    Take a step with momentum: $v^{(i)} \leftarrow \rho \cdot v^{(i-1)} + \nabla^{(i)}, w^{(i)} \leftarrow w^{(i-1)} - \eta v^{(i)}$

**end for**

**Output:** $(\varepsilon_f + \varepsilon_0 + \varepsilon_1)$-Private linear model $w$

---

## 2  A New Linear Scaling Rule

In this section we detail how our method chooses each hyperparameter in DP-SGD, prove the privacy guarantee of the overall hyperparameter selection process, and provide a theoretical analysis.

**A new linear scaling rule**  The well-known linear scaling rule [29] proposes increasing the learning rate with the batch size. We propose a new linear scaling rule that details how to select all hyperparameters in DP-SGD. Our method first fixes full-batch, unit clipping norm, zero initialization and use SGD with momentum, and then jointly scales the learning rate and number of steps with $\varepsilon$. We provide extensive ablations of each design choice in our hyperparameter optimization algorithm in Appendix A.2. Prior work has exclusively taken small step sizes [51; 52; 7; 15; 9] on the order of $\{10^{-5}, 10^{-3}\}$ and works that train transformers have also trained for a small number of epochs $\{1, 3\}$ [51; 7]. While this works well to recover the bulk of the non-private performance when $\varepsilon$ is very small, it is natural to expect that as $\varepsilon \to \infty$ we should increase the parameters of training to more closely resemble that of non-private training. In line with this insight, we propose a linear scaling rule: jointly increase the step size and number of steps linearly with $\varepsilon$. We make use of this simple yet powerful heuristic in the hyperparameter selection strategy that we use in all our experiments, outlined in Algorithm 1. Given a total privacy budget $\varepsilon$, we use an initial portion of this budget to do binary search (random search and grid search are also valid) on the meta-hyperparameter $r = \eta \times T$ for a small value of $\varepsilon$, and use this to estimate the best value of $r$ for the desired overall privacy budget. We provide a privacy guarantee in 2. We note that linear scaling does not hold up forever: we are primarily interested with analyzing $\varepsilon \leq 1$, and show that in this range it holds (Fig. 3).

**Linear Scaling is intuitive.**  Applying the linear scaling rule improves the cosine similarity between noisy weight updates and the optimal solution without degrading accuracy. First note that the classification accuracy of a linear model is scale-invariant; the optimal solution of Gradient Descent with total step size $r$ is $w' = w^*/\|w^*\| \times r$: the projection of $w^*$ onto $B_r$, the ball of radius $r$, and for linear models, the performance (top-1 accuracy) of $w'$ is the same as the performance of $w^*$: $\mathrm{Pred}(w'(x)) = \mathrm{Pred}(w^*(x)) \, \forall x \in D$. An important factor in the success of optimization is the angle between the gradient update $\nabla_i$ and $w'$: if all our updates point in the same direction, we can expect fast convergence. Let similarity(i) $= \frac{\nabla_i \cdot w'}{\|\nabla_i\| \cdot \|w'\|}$. Suppose that $\|w_i\| = \|w'\| \ll 1$, then adding Gaussian noise $\sigma\xi$ where $\xi \sim \mathcal{N}(0, 1)$ to the update will significantly decrease the cosine similarity of the updated model and $w'$. If we decrease $\sigma$, it is easy to see that this mitigates the impact on the trajectory. However, we can equivalently keep $\sigma$ constant and increase the scale of the parameters, and also decrease the impact of noise on the trajectory: similarity($w_i + \sigma\xi, w'$) $<$ similarity($\alpha \cdot w_i + \sigma\xi, \alpha \cdot w'$), $\forall \alpha > 1$. Note that by increasing $r$ we scale the optimal solution

while keeping its performance identical, and thus optimize the cosine similarity of the noisy update. Increasing the number of iterations and the learning rate linearly increases $r$ but does not linearly increase $\sigma$ due to the composition of Gaussian differential privacy [27], therefore the impact on the optimization trajectory is minimized.

**Theory** We introduce two theoretical results. We first analyze the privacy cost including hyperparameter tuning of DP-RAFT under Gaussian DP (GDP). In Thm. 2.3 we analyze the performance gap between hyperparameters for noisy gradient descent in terms of an upper bound in expectation on the distance between private and non-private iterates, and find that applying the linear scaling rule improves the upper bound on this distance. Proofs of all results are in Appendix A.5.

**Proposition 2.1.** *Algorithm 1 is $(\sqrt{T}/\sigma)$-GDP. Moreover, repeating Algorithm 1 for $n$ times for hyper parameter search would be $(\sqrt{T \cdot n}/\sigma)$-GDP.*

**Corollary 2.2.** *Algorithm 1 is $(\epsilon, \Phi(-\epsilon \cdot \sigma/\sqrt{T} + \sqrt{T}/2\sigma)) - e^\epsilon \cdot \Phi(-\epsilon \cdot \sigma/\sqrt{T} - \sqrt{T}/2\sigma))$-DP. Also, for $n$-fold repetition, the algorithm is $(\epsilon, \Phi(-\epsilon \cdot \sigma/\sqrt{n \cdot T} + \sqrt{n \cdot T}/2\sigma)) - e^\epsilon \cdot \Phi(-\epsilon \cdot \sigma/\sqrt{n \cdot T} - \sqrt{n \cdot T}/2\sigma))$-DP*

**Theorem 2.3.** *Let $f$ be gradient descent that minimizes a $\alpha$-strongly convex and $\beta$-smooth function $\ell$ with constant learning rate $\eta \in (0, \frac{2}{\beta})$ over $T$ iterations. Then we can bound the "noisy radius" distance between the noisy iterate $w^T$ and the benign iterate $w_b^T$ at iteration $T$ in expectation:*
$$\mathbb{E}[\|w^T - w_b^T\|] \leq \rho\eta \times (\textstyle\sum_i^{T-1} \max(|1 - \eta\alpha|, |1 - \eta\beta|)^i).$$

Thm. 2.3 indicates that the distance between the noisy and non-noisy weights grows in a very controlled manner; at each iteration the divergence from the previous iteration is decreased by a factor strictly less than 1, and then we add some noise. The main idea of the proof is similar to the main result in Fang et al. [23] but is simpler because we only prove the result for linear models.

We apply this theorem to logistic regression (fine-tuning a linear model on extracted features). In this setting our theorem provides an upper bound on the radius of the range of solutions that DP-SGD produces. For linear models, this radius converts directly into an upper bound on the generalization error. If we use the linear scaling rule to scale $r = \eta \times T$ with $\varepsilon$, we expect that $\eta$ remains appropriately bounded and $T$ does not grow so large that the resulting noise creates significant model drift. Therefore, we find that increasing the quantity $r = \eta \times T$ improves this upper bound.

While our theorem only holds for linear models, we will show that it holds empirically for the deep GPT2 and RoBERTa models, in line with Li et al. [47] who find that even the updates of a large model lie in a low-dimensional space during fine-tuning.

# 3 Evaluation

We provide results on a range of image classification, distribution shift, and natural language processing tasks. Full results for all datasets and models can be found in Appendix A, including ablations on all steps of our method( A.2) and key hyperparameters( A.4).

**Datasets.** We evaluate the performance of our method on 20 benchmark tasks spanning the data modalities of CV and NLP. Image classification: ImageNet [16], CIFAR10, CIFAR100 [40], Fashion-MNIST [80], STL10 [11], EMNIST [12]. Because these image classification datasets are generally considered in-distribution of the pretraining data, we also provide results on a number of distribution shift datasets from the WILDS suite [38] that have been used to evaluate various fine-tuning techniques. CIFAR10 → STL, CIFAR10p1, CIFAR10C, CIFAR100 → CIFAR100C [31], Waterbirds [67], FMoW [10], and Camelyon17 [8]. These datasets are considered benchmark tasks for distribution shifts [42; 43; 53] and include data that is not in-distribution of the training data, making for a more realistic evaluation of the capabilities of our method to solve challenging tasks. We are the first to show that DP-SGD is capable of learning to handle distribution shifts without using any techniques from the distributionally robust optimization (DRO) literature [64]. For NLP tasks we consider text classification tasks from the GLUE benchmark [76]: SST-2, QNLI, QQP, MNLI(m/mm) and for next word generation we use PersonaChat [84], WikiText-2 [54], and Enron Emails [37].

## 3.1 Linear Scaling finds near-optimal hyperparameters with low privacy cost

We first provide a concrete example of the hyperparameter search with $\varepsilon_0$ on CIFAR10. Note that regardless of what strategy we use for hyperparameter search here, our total privacy cost as given by Proposition 2 must be strictly less than $\varepsilon_0$. Binary search, random search, Bayesian optimization and grid search are all methods that we can use for the initial hyperparameter search. For this example, for the sake of simplicity we will use random search with 3 trials, with $\varepsilon_0 = 0.01 \cdot \sqrt{3}, \varepsilon_1 = 0.05 \cdot \sqrt{3}, \varepsilon_f = 0.9, \varepsilon_0 + \varepsilon_1 + \varepsilon_f = 1.0$. For $\varepsilon_0 = 0.01$, we randomly sample r uniformly in the range [1,100]=2,20,100 and then randomly decompose this into (approximate) $(\eta, T)$ pairs of [0.2, 10], [0.5, 40], [1, 100]. These in turn evaluate to accuracies of [91.79, 73.68, 67.21], so the best value of r at $\varepsilon_0 = 0.01$ is 2. We do a similar process at $\varepsilon_1 = 0.05$ and get a best r-value of 5. We do linear interpolation and obtain the line of best fit as $r = 75 \cdot \varepsilon + 1.25$. Approximating this to $r = 75$, we apply the linear scaling rule $r = \eta \times T$ and randomly decomposing this value of $r$ into an $(\eta, T)$ pair of [0.75, 100], we produce a final accuracy of 99.00 at $\varepsilon_f = 0.9$.

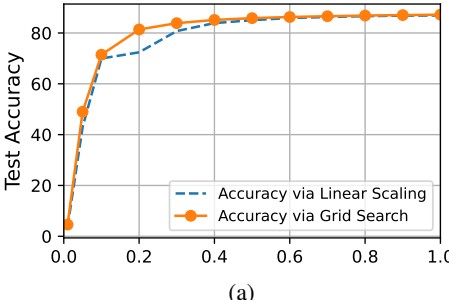

(a)

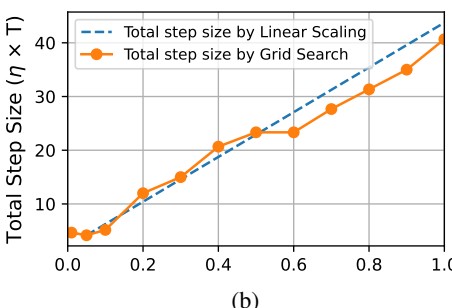

(b)

Figure 3: Training the beit architecture on CIFAR100, the linear scaling rule produces values for $r = \eta \times T$ close to that of grid search, and the performance drop is only apparent at $\varepsilon > 0.2$ because the cost of tuning is $\varepsilon = 0.1$, and vanishingly small for larger $\varepsilon$.

**Linear Scaling outperforms prior hyperparameter search techniques.** We validate the effectiveness of linear scaling against the grid search baseline. In Fig. 3 (right) we compare Alg. 1 to grid search. To avoid scale mismatch on the x-axis we do not account for the privacy cost of grid search, that does $n = 100$ trials (on the same scale as prior work [51]). It is trivial that linear scaling outperforms a naive grid search, but we also compare the effectiveness of linear scaling against the hyperparameter selection strategies used in prior work [51]. We apply linear scaling to the ViT model used in [51] on CIFAR100. Although [51] do not directly state the hyperparameters for their best results, they specify that they use 200 hyperparameter trials with Bayesian optimization. While they obtain RDP guarantees, these guarantees do not include the privacy cost of non-privately tuning hyperparameters. We apply the linear scaling rule to extrapolate a value of $r$ from $\varepsilon = 0.1$ to $\varepsilon = 1$, obtaining $r = 20 = \eta(0.2) \times T(100)$. *We recover performance of 82.7% for $\varepsilon = 1$, a 2% improvement over the best result for DP-Adam in [51] while accounting for the privacy cost of hyperparameter tuning.* They obtain their best result for DP-Adam at $T = 10$, but we cannot compute the corresponding value of $r$ because they do not provide $\eta$. However, because they use a clipping norm of $0.005$ we can reasonably infer that their value of $r$ is $\approx 1000\times$ smaller than ours. This is farther from the optimal non-private training, as evidenced by the performance gap.

Figure 4: Linear Scaling on ImageNet is competitive with prior SOTA [52] (Jan. 2023) and current SOTA [51](within last month).

| $\varepsilon$ | [52] | [51] | Ours | $r = \eta \times T$ |
|---|---|---|---|---|
| 0.25 | 75.6 | - | 79.0 | 250 |
| 0.50 | 79.4 | 86.1 | 81.6 | 750 |
| 1.00 | 81.1 | 86.8 | 83.2 | 1100 |
| 2.00 | 81.5 | 87.4 | 84.2 | 2000 |
| 10.0 | 81.7 | - | 85.4 | 2000 |
| $\infty$ | 86.9 | 88.9 | 85.7 | 2000 |

**Linear Scaling scales to ImageNet** In Table 4 we do a granular comparison between our method and [52; 51]. We observe that our method is competitive with [51] even when accounting for the privacy cost of hyperparameter search, and that the linear scaling rule holds up at the scale of ImageNet for very large values of $r = \eta \times T$. The non-private accuracy of their closed-source model is $3.2\%$ higher than our open-source model, and so the private accuracy at $\varepsilon = 2$ is also $3.2\%$ higher.

However, ultimately our method and the method of Mehta et al. [51] are complementary, because their method introduces new hyperparameters that we intuit our linear scaling rule can optimize. We attempted to validate this intuition empirically but were unable to reproduce the results of Mehta et al. [51] because they and Mehta et al. [52] pretrain on the closed-source JFT dataset with billions of images. We note that all numbers we report for models pretrained on ImageNet-21k using first-order methods surpass those in [51], but for sufficiently small values of $\varepsilon$ on harder datasets the second-order methods they propose provide better performance. We note that the method in Mehta et al. [51] only works for vision tasks, whereas our approach works for both vision and language tasks.

**Linear Scaling produces robust results.** In Fig. 3 we report that following Algorithm 1 produces new state-of-the-art results for all values of $\varepsilon$, shown in Table 5. In Appendix A.1 we provide detailed computations of the linear interpolation for multiple datasets and in Appendix A.4 we provide full results across the entire hyperparameter search space. Our results validate that this rule is robust: we can move from one set of hyperparameters to another similarly performing set of hyperparameters by increasing the number of iterations $T$ by a constant factor and decreasing the learning rate $\eta$ by the same factor (or vice versa). We find that any inaccuracy incurred by estimating the best value of $r$ with the linear scaling rule will not reduce accuracy by much compared to doing grid search for the optimal value of $r$, but does reduce the privacy cost of hyperparameter tuning immensely.

## 3.2 Linear Scaling enables empirical analysis

Many interesting questions in DP fine-tuning remain unanswered because of the immense computational overhead of evaluating hundreds of hyperparameter trials for each privacy budget, model architecture and dataset [51]. We now employ the linear scaling rule to efficiently answer key questions in DP fine-tuning for vision tasks.

**Impact of model architectures on differential privacy** Many pretrained model architectures are available [79] but prior work has generally engaged with a single architecture, e.g. beit [7] or ViT [52]. We leverage our method to answer three questions:

- What model architectures can provide good DP classifiers?

- Is the best model task-specific, e.g., is an architecture search required?

- Does the private-non private utility gap depend on the model architecture?

We report our findings in Tab. 5. We evaluate multiple transformer architectures in ViT [19], beitv1 [4] and beitv2 [58], as well as the purely convolutional architecture Convnext [48]. We find that all architectures can serve as good backbones for high-accuracy DP classification. This is somewhat surprising because the different inductive biases of transformers and purely convolutional architectures tend to produce differently

Figure 5: We compare the best private and best non-private performances of all models on all datasets. We use the linear scaling rule to scale hyperparameters from $\varepsilon = 0.1$ to $\varepsilon = 1$, so our privacy analysis includes the cost of hyperparameter tuning.

| Model | Dataset | $\varepsilon = 1$ | $\varepsilon = \infty$ | Gap |
|---|---|---|---|---|
| beitv2 | CIFAR10 | 99.00 | 99.00 | **0.00** |
| | CIFAR100 | 89.62 | 91.57 | 1.95 |
| | FMNIST | 91.02 | 91.53 | 0.51 |
| | STL10 | 99.69 | 99.81 | 0.12 |
| | EMNIST | 81.77 | 82.00 | 0.23 |
| convnext | CIFAR10 | 96.75 | 97.22 | 0.47 |
| | CIFAR100 | 83.47 | 86.59 | 3.12 |
| | FMNIST | 90.23 | 91.13 | 0.9 |
| | STL10 | 99.61 | 99.71 | 0.10 |
| | EMNIST | 78.38 | 79.05 | 0.67 |
| beit | CIFAR10 | 98.19 | 98.51 | 0.32 |
| | CIFAR100 | 87.1 | 90.08 | 2.98 |
| | FMNIST | 90.55 | 91.6 | 1.05 |
| | STL10 | 99.62 | 99.78 | 0.16 |
| | EMNIST | 81.48 | 83.25 | 1.77 |
| vit-L | CIFAR10 | 98.29 | 98.44 | 0.40 |
| | CIFAR100 | 86.18 | 89.72 | 3.54 |
| | FMNIST | 90.58 | 91.37 | 0.79 |
| | STL10 | 99.62 | 99.76 | 0.14 |

structured features, but we reason that the noise added by DP will 'smooth out' these decision boundaries regardless of architecture. We note that one architecture, beitv2, performs the best on all benchmarks and also has the highest non-private ImageNet accuracy [78]. We therefore recommend that practitioners do not worry about architecture search when fine-tuning as this can incur further privacy costs, and instead pick the best model available. We are encouraged to report that the private-non private utility gap diminishes with model accuracy, enabling us to report for the first time *lossless privacy* of 99.0% on CIFAR10 at $\varepsilon = 1$. We expect that as pretrained models become even better, future works may even be able to attain lossless privacy on CIFAR100, that we note remains somewhat challenging for private fine-tuning. We harness these insights for our next analyses.

**Linear Scaling is robust to distribution shifts.** Benchmarking performance on datasets with distribution shifts is increasingly important because real-world problems almost always contain distribution shift between model training and inference [64]. Prior work in distributionally robust optimization (DRO) has addressed this problem by using knowledge of the relative imbalances between groups, but recent work with vision transformers has shown that linear probing can perform well on datasets with distribution shifts [53; 41; 43]. However there is no work

Figure 6: In-distribution (ID) and out-of-distribution (OOD) performance on benchmark distribution shift datasets. Prior work is non-private (citations are in Appendix A.1). We use the linear scaling rule to scale hyperparameters from $\varepsilon = 0.1$ to $\varepsilon = 1$, so our privacy analysis includes the cost of hyperparameter tuning.

| Dataset | $\varepsilon = 1.0$ ID(OOD) | Prior ($\varepsilon = \infty$) |
|---------|------------------------------|-------------------------------|
| Waterbirds | 92.31 (91.59) | 98.3(80.4) |
| fMoW | 45.44 (35.31) | 49.1 (36.6) |
| Camelyon | 93.91 (93.55) | 99.5 (96.5) |
| C10 → STL | 99.0 (98.82) | 97.5 (90.7) |
| C10 → C10p1 | 99.0 (97.85) | 97.5 (93.5) |
| C10 → C10C | 99.0 (89.98) | 96.56 (92.78) |
| C100 → C100C | 89.65 (68.69) | 81.16 (72.06) |

that evaluates the robustness of private models to distribution shifts. We leverage our method to answer three questions:

- Can DP help when there is a domain shift from private fine-tuning to test?
- Can DP help when there is a domain shift from public data to private fine-tuning?
- Can DP fine-tuned models perform well in the zero-shot setting?

In Table 6 we compare the performance of our method across 8 benchmarks and find that the answer to all three of these questions is *yes*.

The Waterbirds dataset is a well-known benchmark for evaluating the robustness of models to spurious correlations. There is a domain shift between the private training data and the private test data created by class imbalance. We are surprised to find that in the absence of any other regularization methods, DP fine-tuning actually *improves* performance on the OOD split. We hypothesize that the lackluster OOD non-private performance is caused by the model overfitting to the spurious correlation in the training data, and that the inherent regularization of DP prevents the model from memorizing this spurious correlation. By comparing our results to Mehta et al. [53] we determine that this robustness is unique to DP rather than an artifact of the pretrained model. Although DP does significantly degrade the ID performance, in situations where minimizing OOD error is more important, we believe that DP by itself can mitigate the domain shift from private fine-tuning to test.

Because our central assumption in DP fine-tuning is that there is no privacy leakage from the pretraining data to the private training data, it is important to understand how DP fine-tuning performs when there is a distribution shift between public data and private data. fMoW [10] and Camelyon17 [8] are two datasets that represent a signficant distribution from the pretraining data (ImageNet). We observe a similar relationship between ID and OOD degradation as above, where the OOD degradation is somewhat mitigated by DP. If we compare our results on Camelyon to the best results in Ghalebikesabi et al. [25] we find that we can improve their best performance from $91.1\%$ at $\varepsilon = 10$ to $93.91\%$ at $\varepsilon = 1$. Although performance on fMoW remains quite poor, we note that it is not significantly worse than in the non-private setting. We believe that DP fine-tuning from pretrained models remains a viable strategy even when the publicly available pretraining data has a very large distribution shift from the private target data.

We finally consider the zero-shot setting, where we fine-tune a model on CIFAR and then transfer it without updating any parameters to private test datasets that once again represent a distribution shift from CIFAR. We report the performance in the OOD column. For the more minute distribution shifts of STL and CIFAR10p1 we find that the fine-tuned classifier can achieve remarkable performance without ever updating parameters on these datasets; that is, we just remap the labels as per [42]. CIFAR10C and CIFAR100C represent larger distribution shifts and are used to benchmark the robustness of models to commonly reported image corruptions [31]. Our OOD performance on these larger distribution shifts is much worse, particularly for CIFAR100 where there is a $> 20\%$ degradation. Although this is lower than the top result on the RobustBench leaderboard [13] obtains $85\%$ accuracy, we note that once again *we used no additional methods beyond DP to ensure robustness but managed to achieve reasonable performance to distribution shifts in zero-shot classification.*

### 3.3 Linear Scaling for language modeling

Prior work has generally focused on either CV or NLP because the methods used in DP fine-tuning differ greatly across data modalities [46; 51]; here we show that our method extends to NLP by validating on text classification and language modeling tasks. We also update all parameters when fine-tuning, displaying that our method works for both linear probing and full fine-tuning. We fine-tune GPT-2 [63] with our method for three language modeling tasks that have been benchmarked in prior works [46; 70; 30] on private fine-tuning: Persona-Chat [85], WikiText-2 [54] and Enron Emails [37]. We also fine-tune RoBERTa-base on four tasks in the GLUE benchmark: SST-2, QNLI, QQP and MNLI(m/mm) in Table 7.

Figure 7: Linear scaling holds for GLUE tasks when training the full RoBERTa-base model

| Task | $\varepsilon$ | Acc | $r = \eta \times T$ |
|---|---|---|---|
| SST-2 | 0.1 | 90.60 | 0.975 |
| | 0.2 | 90.83 | 1.95 |
| | 1.0 | 91.51 | 9.75 |
| QNLI | 0.1 | 82.54 | 3.9 |
| | 0.2 | 84.00 | 4.68 |
| | 1.0 | 86.25 | 26.52 |
| QQP | 0.1 | 81.07 | 11.7 |
| | 0.2 | 82.21 | 17.55 |
| | 1.0 | 84.69 | 64.35 |
| MNLI(m/mm) | 0.1 | 77.52(78.24) | 11.7 |
| | 0.2 | 79.40(79.98) | 17.55 |
| | 1.0 | 81.86(82.76) | 64.35 |

While prior works mainly focus on $\varepsilon$ in $\{3, 8\}$, in this work we are also interested in smaller $\varepsilon$s like 0.1. Appendix B.1 includes the details for the experimental set-up.

**Linear scaling holds for NLP tasks** We analyze the performance gap between estimated total step size and optimal total step size by grid search to understand how well linear scaling performs on language modeling tasks. Fig. 8 plots the optimal perplexity and perplexity by estimated total step size at different values of $\varepsilon$ on Enron emails. We can see that the linear scaling rule generalizes well for reported values of $\varepsilon$ and the perplexity by the estimated total step size is close to the optimal perplexity. From Table 7 we can see that linear scaling also holds across a range of tasks in the GLUE benchmark. We also have the result for WikiText-2 in Appendix B.3.

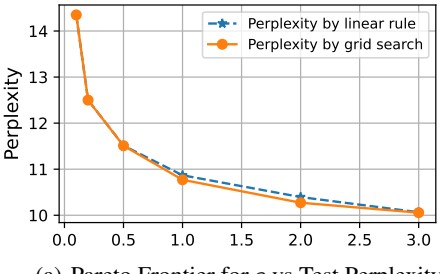

(a) Pareto Frontier for $\varepsilon$ vs Test Perplexity.

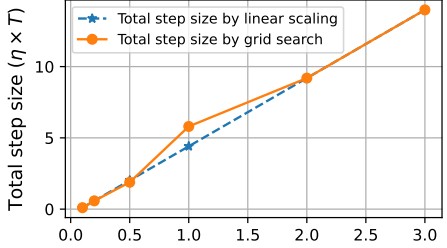

(b) Pareto Frontier for $\varepsilon$ vs Total Step Size.

Figure 8: The linear scaling rule (accounting for the privacy cost of hyperparameter tuning) is competitive with grid search (non-private, doing N trials each with the given $\varepsilon$) on the Enron Emails dataset. Left: y-axis is Perplexity (lower is better).

**The linear scaling rule outperforms prior results on differentially private language modeling tasks.** We first run a qualitative evaluation on the previous benchmark SOTA [46] on PersonaChat trained with DP-SGD by following the linear scaling rule to increase the number of epochs.

Figure 9: Linear scaling holds when fine-tuning all layers of GPT2 on PersonaChat and outperforms Li et al. [46]

| $\varepsilon$ ($\delta = \frac{1}{2|D_{\text{train}}|}$) | 1 | 3 | $\infty$ |
|---|---|---|---|
| Li et al. [46] | - | 24.59 | 18.52 |
| Our Work | 21.25 | - | 17.69 |

We can see in Table 9 that we can push the perplexity under 18 for $\varepsilon = 3$ and $\varepsilon = 8$; this performance is competitive with the non-private baseline. Furthermore, even when pushing for a stricter privacy guarantee $\varepsilon = 0.5$, we can still get perplexity of 21.25, that is better than the result of $\varepsilon = 8$ in [46]. We also report the results of ablating these hyper-parameters and varying the number of layers trained in Appendix B.2.

We quantitatively validate the linear scaling rule on WikiText-2 and Enron email dataset and report the result in Table 10 respectively. We select training parameters and the total step size with Alg. 1.

For WikiText-2, a key observation is that when we compare our results to the best prior reported results in [70], for the same number of passes over the training data (20), we obtain lower perplexity for $\varepsilon = 0.2$ than they report for $\varepsilon = 3$. That is, by just increasing the effective step size from $\sim 8 \times 10^{-6}$ to $\sim 8 \times 10^{-3}$ we can strengthen the privacy guarantee without degrading performance.

Figure 10: Finetuning GPT-2 on WikiText-2 ($\delta = 10^{-6}$) and Enron ($\delta = \frac{1}{2|D_{\text{train}}|}$) with DP-SGD. Ppl is perplexity and TSS is Total Step Size. ($^*$ means estimated). Previously reported best perplexity of GPT-2 on WikiText-2 at $\varepsilon = 3$ is 28.84 in [70].

| Dataset | $\varepsilon$ | 0.1 | 0.2 | 0.5 | 1.0 | 2.0 | 3.0 |
|---------|---|---|---|---|---|---|---|
| WikiText-2 | Ppl | - | 28.81 | 28.37 | 28.15 | 27.98 | 27.69 |
| | TSS | - | 0.008 | 0.02 | 0.04$^*$ | 0.08$^*$ | 0.12$^*$ |
| Enron | Ppl | 14.35 | 12.50 | 11.56 | 10.91 | 10.45 | 10.14 |
| | TSS | 0.10 | 0.58 | 2.02$^*$ | 4.41$^*$ | 9.19$^*$ | 13.98$^*$ |

## 4   Related Work and Discussion

De et al. [15] and Cattan et al. [9] propose the use of large batch sizes and initializing the weights to small values near-zero to standardize training. However, they use ResNet architectures rather than modern vision transformers, and in Appendix A.2 we find that other techniques that they use such as data augmentation, fine-tuning the embedding layer, and weight averaging do not always improve performance. [7] do end-to-end training of the same beit architecture we use, but we crucially observe that updating all parameters incurs the curse of dimensionality and therefore it is better to only update the last layer. Besides vision tasks, Li et al. [46] and Yu et al. [82] provide methods for fine-tuning large language models under DP-SGD by proposing new clipping methods to mitigate the memory burden of per-sample gradient clipping. However, they do not achieve performance comparable to non-private models when fine-tuning a pretrained model on the PersonaChat dataset. We adapt their techniques to the hyperparameter settings that we show are optimal for DP fine-tuning, and produce similar performance to non-private fine-tuning on the PersonaChat dataset. Yu et al. [83] report compelling results by only updating a sparse subset of the LLMs with LoRA [33]. We fine-tune GPT2 and RoBeRTA; Basu et al. [5] also fine-tune BERT models.

Papernot and Steinke [57] propose an RDP hyperparameter optimization algorithm that requires selecting the number of trials at random with a random variable, and exhibits the greatest savings when the number of hyperparameter trials is large. By contrast our linear scaling rule needs only a small fraction of the overall privacy budget for hyperparameter search. Their evaluation only tunes the learning rate of a 3-layer CNN on MNIST. Our rule accounts for multiple hyperparameters (batch size, clipping norm, momentum, learning rate, number of iterations) and produces SOTA results.

Golatkar et al. [26]; Nasr et al. [55]; Amid et al. [2] treat $< 10\%$ of the private training dataset and public and use it to improve DP-SGD. Although we do not use any private data during pretraining, future work can tackle applying linear scaling to this alternate threat model.

An open challenge in DP training is how to privately and efficiently do hyperparameter tuning. We complement the existing body of work by introducing a new linear scaling rule to privately optimize hyperparameters. Our key insight is that we can interpolate between the early-stopping regime that is best for small $\varepsilon$ and the regime of many iterations that is best for $\varepsilon \to \infty$ as $\varepsilon$ increases. We provide find that our method attains new state-of-the-art accuracy across 20 tasks, on benchmark image classification tasks, distribution shift datasets, and natural language modeling tasks.

## 5   Limitations

**Assumptions.** The key assumption in DP fine-tuning is that there is no privacy leakage between public data and private data. We take steps towards qualifying this assumption by evaluating on datasets with distribution shifts between public and private data. **Scope of Claims.** We evaluate 20 datasets across multiple data modalities with multiple model architectures for two types of fine-tuning methods, linear probing and end-to-end training of deep ($> 100M$ param) transformers. **Key Factors that Influence the Performance of Our Approach.** The key parameter in the linear scaling rule is how to allocate privacy budget to the initial hyperparameter search. We find that with privacy budgets as small as $\varepsilon = 0.01$ we can still effectively forecast the linear trend to determine the best hyperparameters for the main privacy budget we consider $\varepsilon = 1$. However, if we need to consider even smaller privacy budgets, it may be challenging to accurately extrapolate hyperparameters.

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
