# A  Further Results for Computer Vision Tasks

Our code is available at the following URL: https://anonymous.4open.science/r/dp-custom-32B9/README.md

## A.1  Experimental Set-up

**Models.**  We evaluate five models: two masked-image modeling transformers, beit [4] and beitv2 [57], their backbone architecture ViT [19] at both the base and large scales, and the pure convolutional architecture convnext [48]. All models are pretrained on ImageNet-21k [16]. These models span a range of input resolutions: beitv2 (224x224), convnext, vit-base, vit-large (384x384), and beit (512x512) and we upsample images to the necessary input size. For text generation we use GPT-2 [61] at the smallest scale, and RoBERTa-base.

**ImageNet.**  For the ImageNet experiments we use a ViT-g that was pretrained on laion-2b, to compare to the ViT-g models that were pretrained on JFT-4b.

**Availability.**  Our results tune open source models from the PyTorch timm package [76] using existing privacy accounting from [27] and per-sample clipping code in [79], and can be reproduced in minutes.

## A.2  Detailed Ablations

Table 1: Improvements obtained by following our method for Private Fine-Tuning on CIFAR100 at $\varepsilon = 0.1$. Details for ablations can be found in Appendix A.2. Across each design choice, we further push the boundaries to improve upon the baseline accuracy in previous works.

| Method | Baseline | Baseline Accuracy | Improvement |
|---|---|---|---|
| Classifier (no bias) | [51] | 71.3 | 0.36 |
| Zero Initialization | Random Initialization [15] | 64.85 | 6.81 |
| Gradient Descent | SGD(Batch=4096) [15] | 70.2 | 1.46 |
| Momentum ($\rho = 0.9$) | $\rho = 0$ [7] | 69.02 | 2.09 |
| PLV Accounting | RDP [15] | 68.43 | 3.23 |
| Unit Clipping ($C = 1$) | $C \ll 1$ [50] | 71.2 | 0.46 |
| Free Step | N/A | 71.11 | 0.55 |

In this subsection we deal with detailed ablations of each step in the method that we use. We ablate each step and show their individual benefits in Table 1. At a high level, we want to maximize the signal-to-noise ratio of updates, accelerate training to minimize the impact of noise on the optimization trajectory, and apply the linear scaling rule to select the best hyperparameters while maintaining a given overall privacy budget.

**1) Extract features from a private dataset using an open source feature extractor pretrained on a public dataset.** A valid criticism of this approach in private fine-tuning is that the fine-tuning dataset can be in-distribution with the training dataset, and this may violate privacy. To address this we evaluate our method on eight datasets that have been used as distribution shift benchmarks in Sec. 3.

**2) Zero-initialize a linear classifier that maps features to classes.** Prior work has studied full network fine-tuning [9; 7; 15] but we find that by doing logistic regression on a linear classifier we minimize the number of parameters, and mitigate the curse of dimensionality. We further simplify the choice of initialization by initializing all parameters to zero.

**3) Apply linear scaling to privately select the step size and number of steps.** We propose a new linear scaling rule: increase either the step size $\eta$ or number of steps $T$ so that the total step size $r = \eta \times T$ is linear in $\varepsilon$. This reduces the hyperparameter search to a binary search in $r$. Furthermore we can do a hyperparameter search for $r$ using a small privacy budget, and then linearly scale up this value to minimize the cost of hyperparameter search(Alg. 1). Using privacy loss accounting enables us to get competitive accuracy for privacy budgets as small as $\varepsilon = 0.01$, so these low-cost trials can

672 inform better hyperparameters. our method already minimizes the private-nonprivate performance
673 gap at $\varepsilon = 1.0$ as we show in Table 2, so spending $\varepsilon = 0.1$ for hyperparameter tuning does not
674 significantly degrade accuracy. Unless stated explicitly otherwise, all privacy-utility tradeoffs reported
675 for our method in the main body include the privacy cost of hyperparameter tuning via the linear
676 scaling rule.

677 **4) Compute the full batch gradient.** This optimizes the signal-to-noise ratio of the update and
678 enables use of large step sizes [28]. We achieve 91.52% accuracy on CIFAR10 ($|D| = 5e4$) for
679 $\varepsilon = 0.01$ when training for 100 epochs with noise multiplier $\sigma = 2561$. When the noise is divided by
680 the batch size, the effective noise multiplier is $\frac{\sigma}{|B|=5e4} \approx 0.05$ and the SNR is $\frac{1}{0.05} = 20$. When we
681 use subsampling with sampling probability $p = 0.2$ and train for the same number of epochs under
682 the same privacy budget, our effective noise multiplier is $\frac{\sigma}{|B|} = \frac{1145}{1e4} = 0.114$, and the corresponding
683 SNR of $\frac{1}{0.114} = 8.7$ is much worse than in the full batch setting.

684 **5) Clip per-sample gradients to unit norm.** As per Eq. 1 reducing the per-sample gradient below 1
685 is equivalent to reducing $\eta$ (and thus reducing the step size) while simultaneously biasing optimization.
686 By setting $c = 1$ we can simplify $r = \eta \times T \times c$ to $r = \eta \times T$.

687 **6) Use privacy loss variable accounting.** Gopi et al. [27] provides a tool to calibrate Gaussian noise
688 for the given privacy budget and add noise to the gradient: this enables budgeting for small values of
689 $\varepsilon$ without underestimating privacy expenditure.

690 **7) Use momentum.** Acceleration has a host of well-known benefits for optimization and is ubiquitous
691 in non-private optimization [59; 36], but prior work has not always used momentum because it
692 can lead DP-SGD astray when the SNR of updates is low [15]. Because we optimize the SNR of
693 individual updates in (4), we can make use of momentum.

694 **8) Take a final step with the same learning rate in the direction of the momentum buffer.** The
695 momentum is private by post-processing, so this step is private without adding noise. Consider a
696 case where every gradient (and thus the momentum) points in the direction of the optimal solution.
697 Even if the last non-private step would achieve 0 loss, the added noise will take the model out of the
698 solution. Taking a step in the direction of the momentum buffer mitigates this, because we do not
699 need to add any noise.

700 **Selecting the Best Model and Training Schedule is Challenging.** There are hundreds of open
701 source models pretrained on ImageNet that can be used as feature extractors, and choosing the best
702 model for the downstream task is critical [39]. A straightforward baseline is to always pick the model
703 with the highest ImageNet top-1 accuracy. However, as we show in Table 2 this greedy baseline does
704 not select the best model for the task. Hyperparameter selection is another critical problem in private
705 fine-tuning, because hyperparameter tuning costs privacy and naive grid search swiftly burns through
706 even generous privacy budgets [56]. While our recipe does not use many hyperparameters, we still
707 need to specify the number of training iterations $T$ and the learning rate $\eta$. Prior work generally trains
708 for a small number of iterations with a small learning rate [7; 9; 50; 15], but as we show in Fig. 19
709 and Fig. 20 this strategy (corresponding to the top left of the heatmaps) is suboptimal.

Table 2: Comparing four models for DP transfer learning with $\varepsilon = 0.1$, we see that choosing the
model based on the pretrained accuracy does not typically produce the best model for the task.

| Model | Pretraining Accuracy (ImageNet) | CIFAR10 | CIFAR100 | STL10 | FashionMNIST |
|---|---|---|---|---|---|
| convnext-384 | 87.54 | 96.03 | 68.38 | **94.48** | 87.72 |
| vit-384 | 87.08 | 96.84 | 62.22 | 80.15 | 83.65 |
| beitv2-224 | 87.48 | **98.65** | 63.25 | 81.58 | **88.87** |
| beit-512 | 88.60 | 97.74 | **72.39** | 94.1 | 88.1 |

710 **Momentum Accelerates Convergence.** Despite the exhaustive study of the acceleration of gradient
711 descent with momentum done by prior work [71; 59] work on DP-SGD generally eschews the use of
712 a momentum term. A notable exception [50] use AdamW rather than SGD with momentum; in a
713 later section we discuss the reason to prefer SGD with momentum. The reason to use momentum
714 to accelerated the convergence of DP-SGD is straightforward: the exponentially moving average of
715 noisy gradients will have higher SNR than individual gradients. Furthermore, momentum is shown

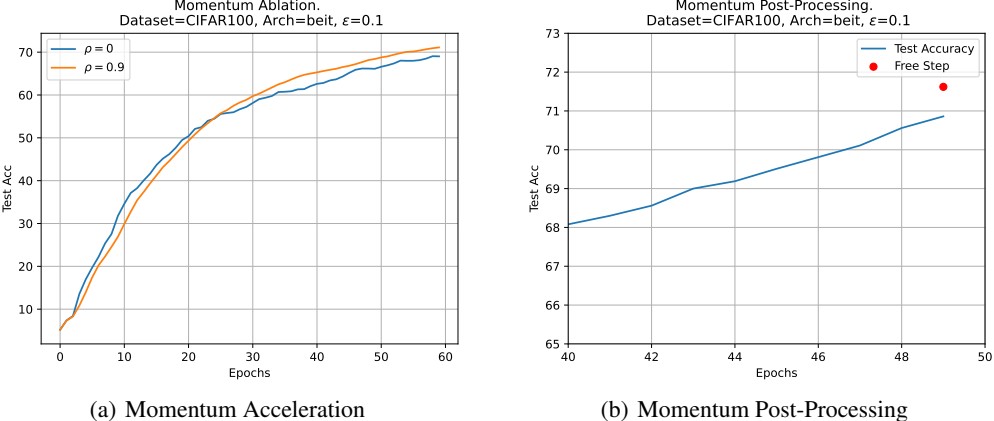

(a) Momentum Acceleration

(b) Momentum Post-Processing

Figure 11: Ablation of momentum parameter during training (left) and post processing of the parameter exponential moving average stored in the momentum buffer to take an extra step 'for free' (right). Use of both methods increases performance slightly.

to provably benefit normalized SGD [14]. In Fig. 11 we observe that momentum complements our new linear scaling rule and accelerates convergence. Separately, we report the improvement of taking a step 'for free' in the direction of the exponential moving average stored during training in the momentum buffer. Note that this exponential moving average is in no way tied to momentum, and it is equivalent to perform DP-SGD without acceleration, store an exponential moving average of gradients with decay parameter $\gamma = 0.9$, and then take an additional step in the direction of the stored gradient average after training has finished; we only use the momentum buffer for ease of implementation. As we discuss above when introducing the new linear scaling rule, we maximize performance by maximizing SNR and terminating training while the model is still improving. Intuitively we therefore expect that the momentum buffer will contain a good estimate of the direction of the next step that we would have taken had we continued training, and taking a step in this direction with our usual learning rate should only improve performance without any privacy loss. We use momentum with $\rho = 0.9$ in all other experiments and also take a 'free step' at the end of private training.

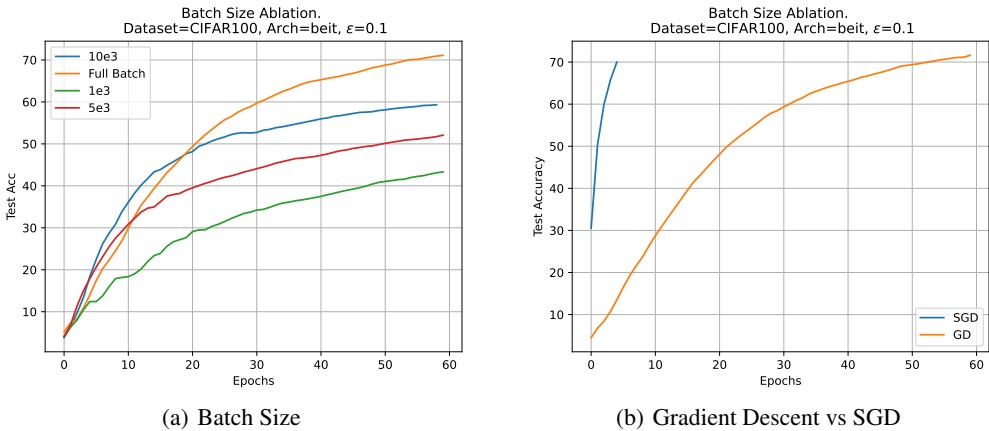

(a) Batch Size

(b) Gradient Descent vs SGD

Figure 12: Ablation of batch size. Left: We vary the batch size using the learning rate and number of iterations tuned for full batch; all other batch sizes perform much worse. Right: We compare SGD and GD. For SGD we tune the batch size jointly with learning rate and number of iterations, arriving at a batch size of 4096 and plot the best performing run against full batch.

**Full Batches Optimize Signal-to-Noise Ratio.** Since its inception, the use of privacy amplification via Poisson subsampling and RDP has been a mainstay in the DP community [84; 75; 22]. Prior work

731 almost universally uses privacy amplification via subsampling, but as early as [49], and more recently
732 in [15] it has become apparent that DP-SGD can actually benefit from large batch sizes because the
733 signal-to-noise ratio (SNR) improves. Note that the noise term in 1 is divided by the batch size, so if
734 we are willing to give up amplification via subsampling entirely, we can reduce the noise by a factor
735 of $5e4$ for the benchmark computer vision tasks. In Fig. 12 we report the improvement of full-batch
736 DP-GD over Poisson subsampled DP-SGD. We attribute the success of DP-GD to the improvement
737 in SNR. For example, we achieve $91.52\%$ accuracy on CIFAR10 for $\varepsilon = 0.01$ when training for 100
738 epochs with learning rate $\eta = 0.01$ and noise multiplier $\sigma = 2561$. When the noise is divided by the
739 batch size, the effective noise multiplier is $\frac{\sigma}{|B|=5e4} = 0.05$ and the SNR is $\frac{1}{0.051} = 20$. When we use
740 subsampling with sampling probability $p = 0.2$ and train for the same number of epochs under the
741 same privacy budget, our effective noise multiplier is $\frac{\sigma}{|B|} = \frac{1145}{1e4} = 0.114$, and the corresponding
742 SNR of $\frac{1}{0.114} = 8.7$ is much worse than in the full batch setting. Although at first glance our analysis
743 merely supports the typical conclusion that large batches are better in DP-SGD, [15] observe that
744 DP-SGD is still preferrable to DP-GD because minibatching produces the optimal choice of noise
745 multiplier. Our findings run counter to this: as discussed above, we contend that performance depends
746 not only on the optimal noise multiplier but on our new linear scaling rule, and DP-GD unlocks the
747 use of larger step sizes [28]. We use DP-GD instead of DP-SGD in all other experiments, removing
748 the batch size from the hyperparameter tuning process and improving the overall privacy cost of
749 deploying our baselines [56].

## A.3   A Critical Evaluation of Proposed Techniques for Fine-Tuning

751 Prior work has proposed a number of ad-hoc techniques that improve performance in DP fine-tuning.
752 Here we critically evaluate these techniques in the our method regime, and analyze why they reduce
753 performance in our setting.

754 **Small Clipping Norms Bias Optimization.**   The standard deviation of the noise added in DP-SGD
755 scales with the sensitivity of the update, defined by the clipping norm parameter. To decrease the
756 amount of noise added, prior work has used very strict clipping [50; 7]. Intuitively, if the clipping
757 norm parameter is already chosen to be some value smaller than the norm of the unclipped gradient,
758 the gradient estimator is no longer unbiased and this may have a negative impact on optimization. In
759 Fig. 14 we observe that decreasing the clipping norm below 1 only degrades performance. As we
760 can see in equation 1, further decreasing the clipping norm is equivalent to training with a smaller
761 learning rate, and this is suboptimal because Fig. 19 indicates that we can prefer to use larger learning
762 rates. We use a clipping norm of 1 in all other experiments.

763 **Initializing Weights to Zero Mitigates Variance in DP-GD.**   [60] propose initializing the model
764 parameters to very small values to improve the stability of micro-batch training, and [15] find that
765 applying this technique to DP-SGD improves performance. In Fig. 13 we ablate the effectiveness
766 of *zero initialization* with standard He initialization and find that the best performance comes from
767 initializing the weights uniformly to zero. We initialize the classifier weights to zero in all other
768 experiments.

769 **Weight Averaging Cannot Catch Up To Accelerated Fine-Tuning.**   [66] perform an in-depth
770 empirical analysis and find that averaging the intermediate model checkpoints reduces the variance of
771 DP-SGD and improves model performance. [15] first proposed the use of an Exponential Moving
772 Average (EMA) to mitigate the noise introduced by DP-SGD. Previously, methods that use stochastic
773 weight averaging (SWA) during SGD have been proposed and are even available by default in
774 PyTorch [34]. The idea of averaging weights to increase acceleration was first proposed by [58],
775 and is theoretically well-founded. In Fig. 15 we compare EMA and SWA with no averaging and
776 find that no averaging performs the best. This is because weight averaging methods work well
777 when optimization has converged and the model is plotting a trajectory that orbits around a local
778 minima in the loss landscape [34]. That is to say, the model's distance from the initialization does not
779 continually increase and at some point stabilizes so that the weight averaging method can 'catch up'.
780 However, as discussed in Fig. 3 the optimal number of iterations for our method is to train for longer
781 epochs without decaying the learning rate for convergence, because when the model converges the
782 SNR decays. This is corroborated by Fig. 15, where we see that the distance from initialization is
783 monotonically increasing. Our findings run counter to those of [66] for hyperparameters in line with

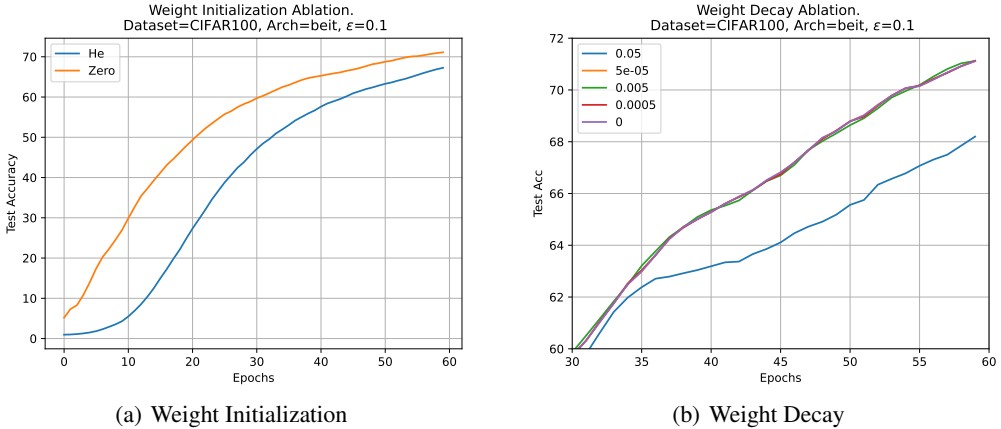

(a) Weight Initialization

(b) Weight Decay

Figure 13: Ablation of two previously proposed methods: zero initialization of parameters and weight decay. Zero initialization increases accuracy in all experiments, but weight decay only degrades performance.

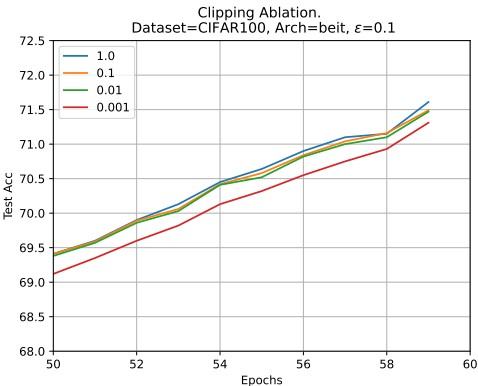

Figure 14: Because reducing the clipping norm is equivalent to reducing the learning rate, reducing the clipping norm below 1 only degrades performance on CIFAR100 for the beit architecture at $\varepsilon = 0.1$.

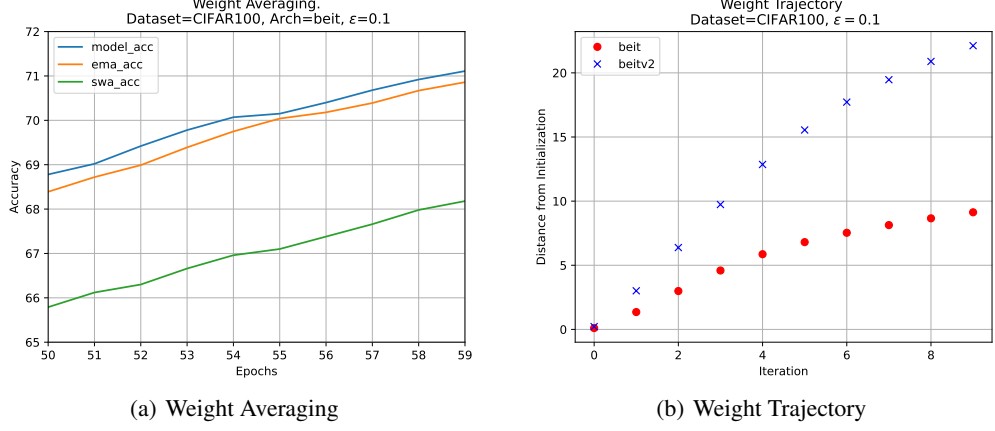

(a) Weight Averaging

(b) Weight Trajectory

Figure 15: Left:Ablation of Weight Averaging. Right: Plot of distance from initialization. Weight Averaging does not improve performance because the model is monotonically moving away from the initialization and weight averaging cannot 'catch up'.

our proposed linear scaling rule because we find that the best optimization regime for our method is precisely one where weight averaging can never catch up to the optimization trajectory. Therefore, the averaging methods only serve to lag one step behind no averaging.

**Data Augmentation Does Not Work When Freezing Embeddings.** Data augmentation is used during training to bias the model towards selecting features that are invariant to the rotations we use in the augmentations. [24] find that feature extractors pretrained on ImageNet are naturally biased towards texture features. [15] eschew traditional data augmentation and instead propose the use of multiple dataset augmentations or "batch augmentation", first introduced by [32], to mitigate the variance of DP-SGD. In Fig. 16 we ablate the effectiveness of batch augmentation and find that it does not noticeably improve accuracy during transfer learning. This is because dataset augmentation changes the prior of the model when training the entire network [69], but when we freeze all layers but the classifier, the model does not have the capacity to change to optimize for the prior introduced by data augmentation, because the embedding layer is frozen.

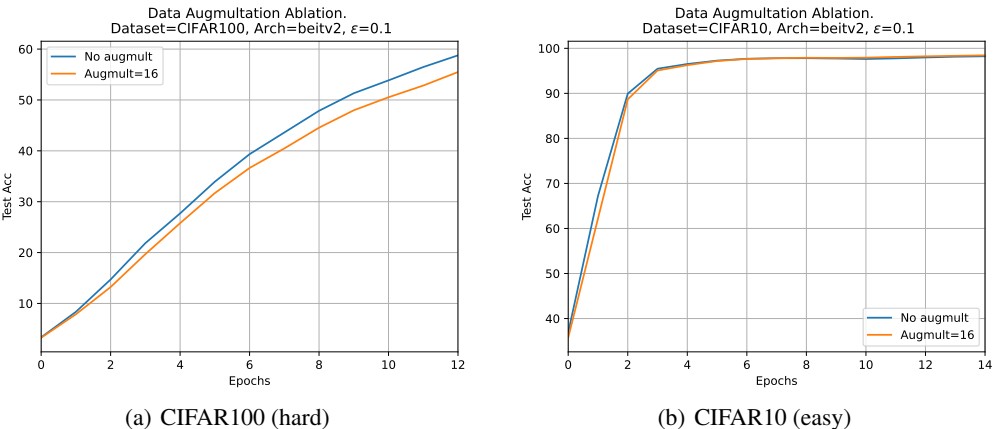

(a) CIFAR100 (hard)  (b) CIFAR10 (easy)

Figure 16: Ablation of Data Augmultation on two datasets. On both datasets, Data Augmultation lags behind the baseline because there is much more training data, and even at the end, Data Augmultation does not have a noticeable improvement.

**Weight Decay Is Not Needed When Freezing Embeddings.** Regularization methods such as weight decay are commonly used during pretraining to prevent overfitting, and the feature extractors we use are pretrained with AdamW [19]. One of the benefits of weight decay during fine-tuning is limiting the change of the embedding layer to not overfit and thus retain the features learned during pretraining [42]. In the ongoing debate on whether to use weight decay during fine-tuning [73], we submit that weight decay should not be used in private fine-tuning. In Fig. 13 we ablate a range of values of the weight decay parameter and observe that increasing the weight decay beyond a negligible amount (the gradient norm is $\approx 1e - 2$) only decreases accuracy, and no value of the weight decay increases accuracy. There are two reasons for this. The first is that we initialize the weights of the model to zero, so we do not expect the gradients to be large. The second is that we only train the last layer, and therefore there is no need to regularize the training of the embedding layer. This supports the conclusion of [43] that SGD with momentum is outperforms AdamW as long as the embedding layer is not updated.

**Details on OOD Experiments** We specify exact details for all OOD experiments. Our training details are drawn from prior work [43; 42; 17]. **Waterbirds**: the ID→OOD contains a well-studied spurious correlation in the binary classification problem. [52] evaluate vision transformers without using group knowledge and obtain $\approx 80\%$ ID accuracy, but much worse ($\approx 60\%$) OOD accuracy, and [43] tailor their method to this task and get the reported results. Surprisingly, just fine-tuning a linear model on the extracted features outperforms both works for OOD accuracy for $\varepsilon = 0.1$. This trend (sacrificing ID accuracy for increased OOD robustness) is seen in other OOD results, and we hypothesize that this is due to the inherent regularization present in DP-SGD.

**Fmow:** we train on region 3 (ID) and evaluate on regions 1,2 (OOD), following [42].

Table 3: We compare the best private and best non-private test accuracy performances of our method to prior work using models pretrained on ImageNet-21k and fine-tuned on CIFAR10 and CIFAR100. Full results are in Section 3.

| Model | Dataset | $\varepsilon = 0.1$ | $\varepsilon = 1$ | $\varepsilon = \infty$ | Gap $(1 - \infty)$ |
|---|---|---|---|---|---|
| our method | CIFAR10 | **98.65** | **99.00** | 99.00 | **0.00** |
| | CIFAR100 | **81.9** | **89.81** | 91.57 | 1.76 |
| [50] | CIFAR10 | 95.8 | 96.3 | 96.6 | 0.3 |
| | CIFAR100 | 78.5 | 82.7 | 85.29 | 2.59 |
| [7] | CIFAR10 | - | 96.7 | 97.4 | 0.7 |
| | CIFAR100 | - | 83.0 | 88.4 | 5.4 |
| [9] | CIFAR10 | - | 95.0 | 96.4 | 1.4 |
| | CIFAR100 | - | 73.7 | 82.1 | 8.4 |
| [15] | CIFAR10 | - | 94.8 | 96.6 | 1.8 |
| | CIFAR100 | - | 67.4 | 81.8 | 14.4 |

**Camelyon17:** we again follow [42].

**CIFAR10 $\rightarrow$ STL10, CIFAR10p1:** We train privately on CIFAR10 using our best hyperparameters returned from the linear scaling rule and then transfer this to STL10/CIFAR10p1, with the label reassignment following [43].

**Common Corruptions:** We evaluate on the average severity of the 'gaussian blur' corruption.

## A.4   Hyperparameter Ablations

We provide full heatmaps and pareto frontiers for all datasets and the 3 best performing models (we do not perform a full evaluation on the ViT in order to minimize any knowledge leak for the evaluation of the linear scaling rule with the strategy in [50]). We note that while all of these datasets are arguably in-distribution, our focus is on comparing the regime of optimization preferred by our method to those of other works, and this is achieved by producing results on benchmark tasks. We further note that STL10 is explicitly in-distribution for the pretraining dataset (ImageNet); we only use this dataset as a temporary stand-in for evaluation on ImageNet-1k, a common benchmark in prior work [50] to minimize the computational burden.

**Hyperparameter Tuning and Selecting Epsilon.**   Prior work often uses unrealistic values of $\varepsilon$ that provide no real privacy guarantee. While some prior work makes the case that hyperparameters need to be tuned even for non-private learning and can be chosen beforehand, we show that this is not the case. Not only are the optimal choices of key hyperparameters different between training from scratch and transfer learning [45], they are also different for non-private and private transfer learning [46; 15]. We now provide guidelines for selecting $\varepsilon$ and broad intuition behind our choice to design a system that minimizes dependence on hyperparameters.

For a decade the standard values of $\varepsilon$ proposed for privacy preserving statistics queries have fallen in the range of 0.1 in line with $e^\varepsilon \approx 1 + \varepsilon$ for $\varepsilon \ll 1$ [20], and recently surveyed DP deployments generally abide by the rule of selecting $\varepsilon \approx 0.1$ [21]. We know that while all small values of $\varepsilon$ generally behave the same, every large value of $\varepsilon$ is fundamentally different in a unique way [21]. In line with these guidelines, we only evaluate $\varepsilon \in [0.01, 1.0]$ and perform most of our ablations on the most challenging task where we can see a range of performance: CIFAR100 for $\varepsilon = 0.1$.

## A.5   Theory

Proof of Proposition 2:

*Proof.*  Since we are using the full batch, each iteration of the algorithm is an instantiation of the Gaussian mechanism with sensitivity of 1 and Gaussian noise with standard deviation of $\sigma$. Hence, each iteration of the mechanism is $(1/\sigma)$-GDP by Theorem 3.7 in [18]. Then, since we have the adaptive composition of $T$ of these mechanisms, the algorithm is $(\sqrt{T}/\sigma)$-GDP overall, using the composition theorem for GDP, as stated in Corollary 3.3 in [18]. $\qquad\qquad\square$

853 Proof of Corollary 2.2:

854 *Proof.* This directly follows from the GDP to DP conversion as stated in Corollary 2.13 in [18]. □

855 Proof of Thm. 2.3

856 *Proof.* We first apply [64] to see that gradient descent with step size $\frac{2}{\beta} > \eta > \frac{2}{\alpha+\beta}$ on a $\alpha$-strongly
857 convex, $\beta$-smooth function is a $\max(1 - \eta\beta, 1 - \eta\alpha)$-contraction. Call this latter quantity $c$.
858 Now consider a sequence of benign updates from gradient descent $w_b^t$ and a sequence of noisy
859 updates for the same dataset $w^t$. Given the contractive property of GD , we have the following:
860

$$\left| (w_b^t - \eta\nabla f(w_b^t)) - (w_v^t - \eta\nabla f(w^{(t)})) \right| \leq c \left| w_b^t - w_b^{t-1} \right| \tag{1}$$

861 We apply the update rule in 1 and use Eq.1

$$w^{(t+1)} = w^{(t)} - \eta(\nabla f(w^{(t)}) + \sigma\xi) \tag{2}$$

$$\left| w_b^{t+1} - w^{t+1} \right| = \tag{3}$$

$$= \left| w_b^t - \eta\nabla f(w_b^t) - w^{(t)} + \eta\nabla f(w^{(t)}) - \sigma\xi \right| \tag{4}$$

$$\leq c \left| w_b^t - w^{(t)} \right| + \eta\rho \tag{5}$$

862 Now we have the following

$$\left| w^t - w_b^t \right| \leq c \left| w^{t-1} - w_b^{t-1} \right| + \rho\eta \tag{6}$$

863 We now proceed via induction. Assume for $T - 1$ the statement of Thm. 2.3 holds. By Eq.6 and the
864 induction hypothesis we have

$$\left| w^{T-1} - w_b^{T-1} \right| \leq \rho\eta \times \left( \sum_i^{T-2} c^i \right) \tag{7}$$

$$\left| w^T - w_b^T \right| \leq c \left( \rho\eta \times \left( \sum_i^{T-2} c^i \right) \right) + \rho\eta \tag{8}$$

$$\left| w^T - w_b^T \right| \leq \rho\eta \times \left( \sum_i^{T-1} c^i \right). \tag{9}$$

$$\rho\eta \times \left( \sum_i^{T-1} c^i \right) = \frac{\rho\eta(1 - c^T)}{1 - c}$$

$$\rho\eta \frac{1 - c^T}{1 - c} = \frac{\rho\eta(1 - c^T)}{\eta \cdot \min(\alpha, \beta)} = \frac{\rho(1 - c^T)}{\min(\alpha, \beta)}$$

865 The intuition is clear: at iteration 0 there is no divergence. At iteration 1 there is $\eta\rho$ divergence. At
866 iteration 2 the previous divergence contracts by $c$ and increases by $\eta\rho$, so the divergence is $c^1\eta\rho + \eta\rho$.
867 At iteration 3 the divergence is $c^2\eta\rho + c^1\eta\rho + \eta\rho = \eta\rho(c^2 + c + 1)$. We refer to the analysis from [55]
868 on the convexity and smoothness, and the resulting constants, for logistic regression. □

869 # B  Furthur Results for Language Modeling Tasks

870 ## B.1  Experimental Set-up for Finetuning Language Models

871 **Persona-Chat:**    We write code based on winners of ConvAI2 competition[1] and private-transformers
872 library.[2] We first do clipping norm $[0.1, 0.2, 0.5, 1.0]$, learning rate in $[2, 5, 10, 20, 50] \times 10^{-5}$, batch

---

[1]https://github.com/huggingface/transfer-learning-conv-ai.
[2]https://github.com/lxuechen/private-transformers.

size 64 and epochs $[3, 10, 20]$ at $\varepsilon = 3$ and $\varepsilon = 8$ and find that the clipping norm in this range achieves almost same perplexity with other hyperparams fixed. We then do hyperparameter tuning as reported in Table 4 to finetune GPT-2.

Table 4: Set of hyper-parameters used in the finetuning GPT-2.

| Parameter | Values |
| --- | --- |
| Clipping Norm | 0.1 |
| Learning Rate | $[2, 5, 10, 20, 50, 100] \times 10^{-5}$ |
| Batch Size | $[64, 128, 256, 512, 1024]$ |
| Epochs | $[3, 10, 20]$ |

**WikiText-2:**  We write code based on the HuggingFace transformers library GPT-2 example,[3] source code by [68][4] and private-transformers library. The hyperparameter range for grid search is reported in Table 5.

Table 5: Set of hyper-parameters for grid search to finetune GPT-2 on WikiText-2. $\delta = 10^{-6}$.

| Parameter | Values |
| --- | --- |
| Clipping Norm | 1 |
| Batch Size | 2048 (Full Batch) |
| Epochs | 20 |
| Learning Rate for $\varepsilon = 0.1$ | $[2, 3, 4, 5, 6, 7, 8, 9, 10, 20] \times 10^{-4}$ |
| Learning Rate for $\varepsilon = 0.2$ | $[2, 3, 4, 5, 6, 7, 8, 9, 10, 20] \times 10^{-4}$ |
| Learning Rate for $\varepsilon = 0.5$ | $[0.7, 0.8, 0.9, 1, 2, 3, 4, 6, 8, 10] \times 10^{-3}$ |
| Learning Rate for $\varepsilon = 1.0$ | $[0.8, 1, 2, 3, 4, 6, 8] \times 10^{-3}$ |
| Learning Rate for $\varepsilon = 2.0$ | $[1, 2, 3, 4, 5, 6, 7, 8, 9, 10] \times 10^{-3}$ |
| Learning Rate for $\varepsilon = 3.0$ | $[0.5, 0.6, 0.7, 0.8, 0.9, 1.0, 1.2, 1.4, 1.6, 1.7, 1.8, 2.0] \times 10^{-2}$ |

**Enron Email:**  For Enron email dataset, we use the preprocessed dataset in [30], where the non-private baseline of finetuned GPT-2 on this dataset is 7.09. The hyperparameter range for grid search is reported in Table 6.

Table 6: Set of hyper-parameters for grid search to finetune GPT-2 on Enron Email dataset. $\delta = \frac{1}{2|D_{\text{train}}|}$.

| Parameter | Values |
| --- | --- |
| Clipping Norm | 1 |
| Batch Size | 1024 |
| Epochs | 5 |
| Learning Rate for $\varepsilon = 0.1$ | $[2, 3, 4, 5, 6, 7, 8, 9, 10] \times 10^{-4}$ |
| Learning Rate for $\varepsilon = 0.2$ | $[0.6, 0.8, 1, 2, 3, 4, 6, 7] \times 10^{-3}$ |
| Learning Rate for $\varepsilon = 0.5$ | $[0.4, 0.6, 0.8, 0.9, 1, 1.1, 1.2, 1.3, 1.4, 1.5, 1.6, 1.8, 2] \times 10^{-2}$ |
| Learning Rate for $\varepsilon = 1.0$ | $[1, 2, 3, 4, 5, 6, 7, 8] \times 10^{-2}$ |
| Learning Rate for $\varepsilon = 2.0$ | $[2, 3, 4, 5, 6, 7, 8, 9, 10] \times 10^{-2}$ |
| Learning Rate for $\varepsilon = 3.0$ | $[0.6, 0.7, 0.8, 0.9, 1.0, 1.1, 1.2, 1.3, 1.4, 1.6, 1.8, 2.0] \times 10^{-1}$ |

### B.2  Additional Results on Persona-Chat

We report the perplexity of GPT-2 on the Persona-Chat dataset at different epochs and batch size in Figure 17 (with tuned learning rate in Table 4) and we can see that larger batch size and longer

---

[3]HuggingFace transformers GPT-2 example code.
[4]https://github.com/wyshi/sdp_transformers

epochs can achieve better perplexity, which is consistent with our linear scale rule. Besides, we also investigate fine-tuning multiple layers. With letting the embedding layer and last LayerNorm layer in transformer trainable, we consider fine-tuning only last block in transformer, first and last block in transformer and report the result in Table 7 and we can see that the best perplexity is achieved by fine-tuning the whole model.

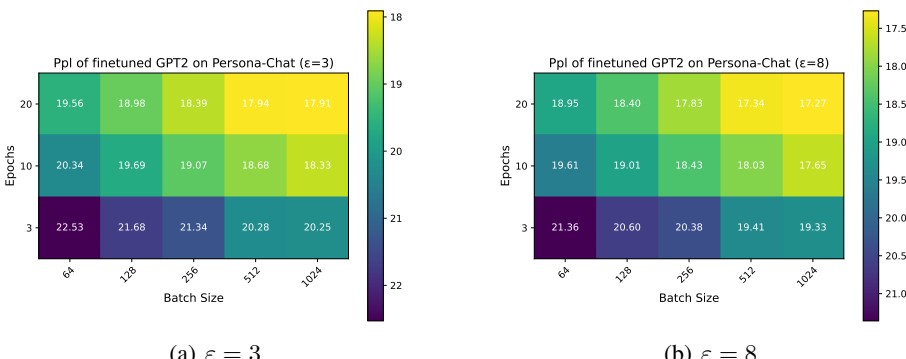

(a) $\varepsilon = 3$          (b) $\varepsilon = 8$

Figure 17: Comparison of perplexity at different batch size and epochs of GPT-2 on Persona-Chat dataset.

Table 7: Finetuning GPT-2 on Persona-Chat dataset including full model and different layers of model. We also include non-private baseline.

| $\varepsilon$ | 3 | 8 |
|---|---|---|
| Full | 17.91 | 17.27 |
| Last Block | 19.80 | 19.20 |
| First-Last-Block | 18.93 | 18.26 |

## B.3   Addtional Results on WikiText-2

We run the grid-search experiment for $\varepsilon \in \{0.2, 0.5, 1, 2, 3\}$ to evaluate the performance gap between the optimal total step size and the estimated total step size.[5]) and present the result in Figure 18. The linear rule scales well from $\varepsilon \in \{0.2, 0.5\}$ to $\varepsilon = 1$. Though for $\varepsilon \in \{2, 3\}$ the perplexity of total step size by linear scale rule is slightly higher than the optimal perplexity of total step size by grid search, the result by linear scale is better than previous SOTA [68], which is $28.84$ at $(\varepsilon = 3, \delta = 10^{-6})$ by training 20 iterations.

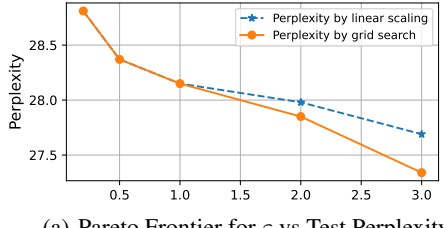 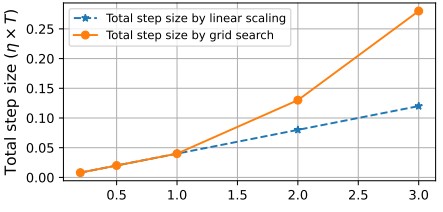

(a) Pareto Frontier for $\varepsilon$ vs Test Perplexity      (b) Pareto Frontier for $\varepsilon$ vs Total Step Size

Figure 18: The linear scaling rule (accounting for the privacy cost of hyperparameter tuning) is competitive with grid search (non-private, doing N trials each with the given $\varepsilon$) in range $[0.2, 1.0]$ on the WikiText-2 dataset. Left: y-axis is Perplexity (lower is better).

---

[5]Due to the limit of computation resources, all experiments are done by training for 20 iterations. Further increasing the number of iterations will help improve the utility as shown by previous study [46; 68], we leave longer iterations for further study.

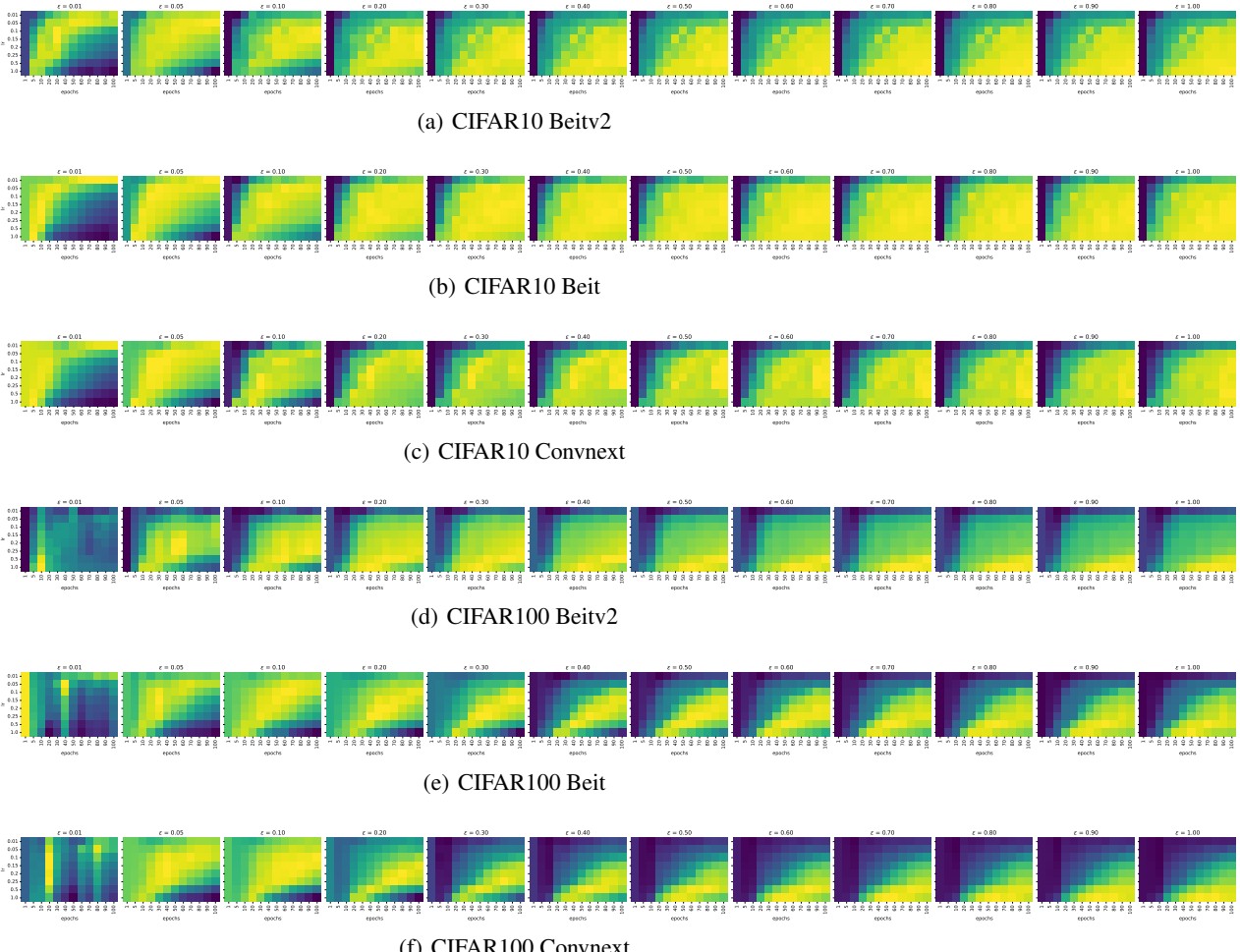

(a) CIFAR10 Beitv2

(b) CIFAR10 Beit

(c) CIFAR10 Convnext

(d) CIFAR100 Beitv2

(e) CIFAR100 Beit

(f) CIFAR100 Convnext

Figure 19: Heatmaps for the reported datasets and architectures; lighter is better. Note that the scale of the axes differs from the heatmaps in the main body; this will be fixed in a future update. $\varepsilon$ increases left to right with a different value for each heatmap according to: $[0.01, 0.05, 0.1, 0.2, 0.3, 0.4, 0.5, 0.6, 0.7, 0.8, 0.9, 1.0]$, epochs increase from left to right on the x-axis of each heatmap according to: $[1, 5, 10, 20, 30, 40, 50, 60, 70, 80, 90, 100]$, and the learning increases from top to bottom on the y-axis of each heatmap according to: $[0.01, 0.05, 0.1, 0.15, 0.2, 0.25, 0.5, 1.0]$. As $\varepsilon$ increases, left to right, the optimal hyperparameters trend towards longer training with lower learning rates (bottom right).

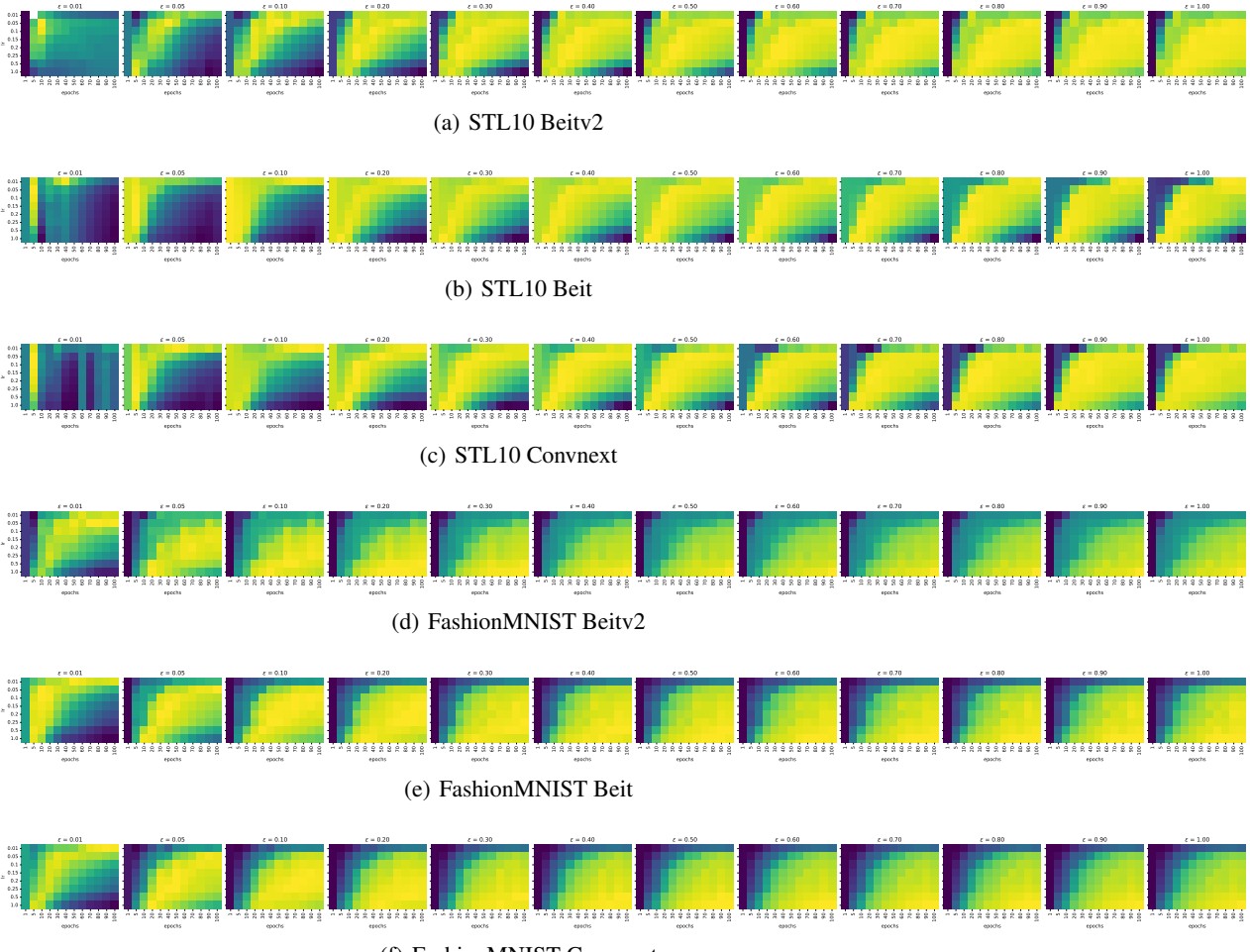

(a) STL10 Beitv2

(b) STL10 Beit

(c) STL10 Convnext

(d) FashionMNIST Beitv2

(e) FashionMNIST Beit

(f) FashionMNIST Convnext

Figure 20: Heatmaps for the reported datasets and architectures; lighter is better. Note that the scale of the axes differs from the heatmaps in the main body; this will be fixed in a future update. $\varepsilon$ increases left to right with a different value for each heatmap according to: $[0.01, 0.05, 0.1, 0.2, 0.3, 0.4, 0.5, 0.6, 0.7, 0.8, 0.9, 1.0]$, epochs increase from left to right on the x-axis of each heatmap according to: $[1, 5, 10, 20, 30, 40, 50, 60, 70, 80, 90, 100]$, and the learning increases from top to bottom on the y-axis of each heatmap according to: $[0.01, 0.05, 0.1, 0.15, 0.2, 0.25, 0.5, 1.0]$. As $\varepsilon$ increases, left to right, the optimal hyperparameters trend towards longer training with lower learning rates (bottom right).

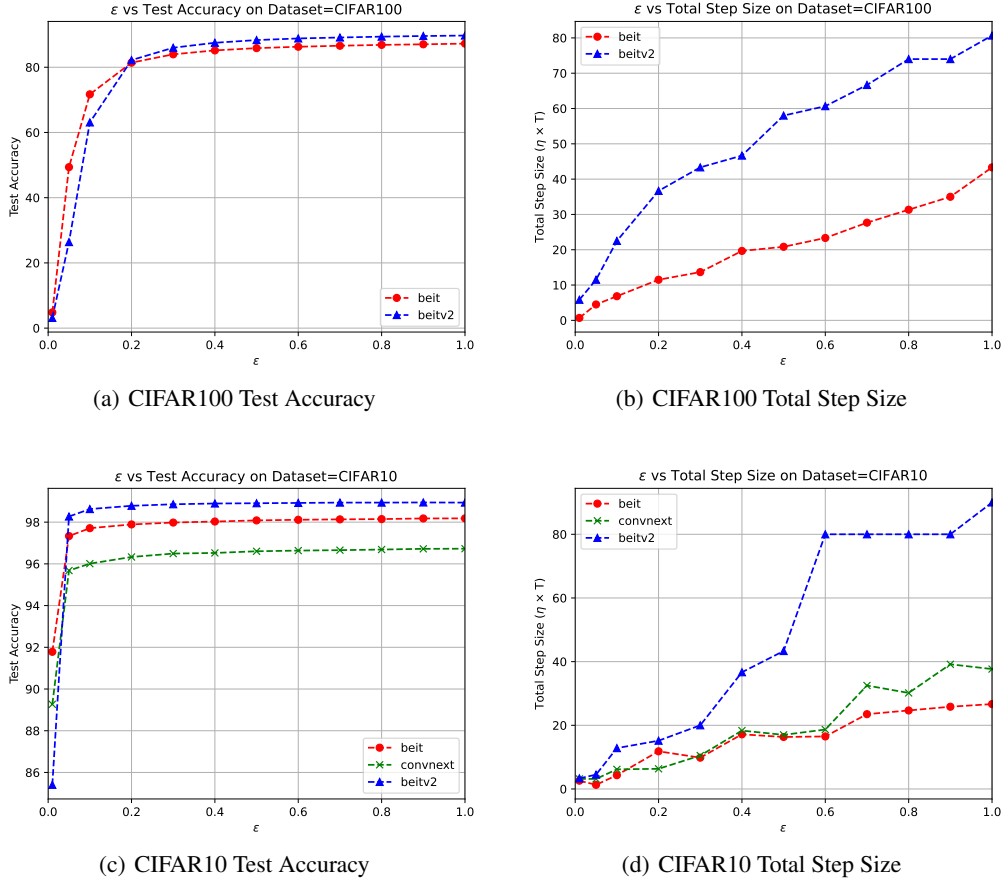

(a) CIFAR100 Test Accuracy

(b) CIFAR100 Total Step Size

(c) CIFAR10 Test Accuracy

(d) CIFAR10 Total Step Size

Figure 21: Pareto frontier for $\varepsilon$ vs test accuracy and total step size for CIFAR10, and CIFAR100. Beitv2 excels for larger values of $\varepsilon$ but beit and convnext are better for smaller values of $\varepsilon$. The inflection point varies across datasets.

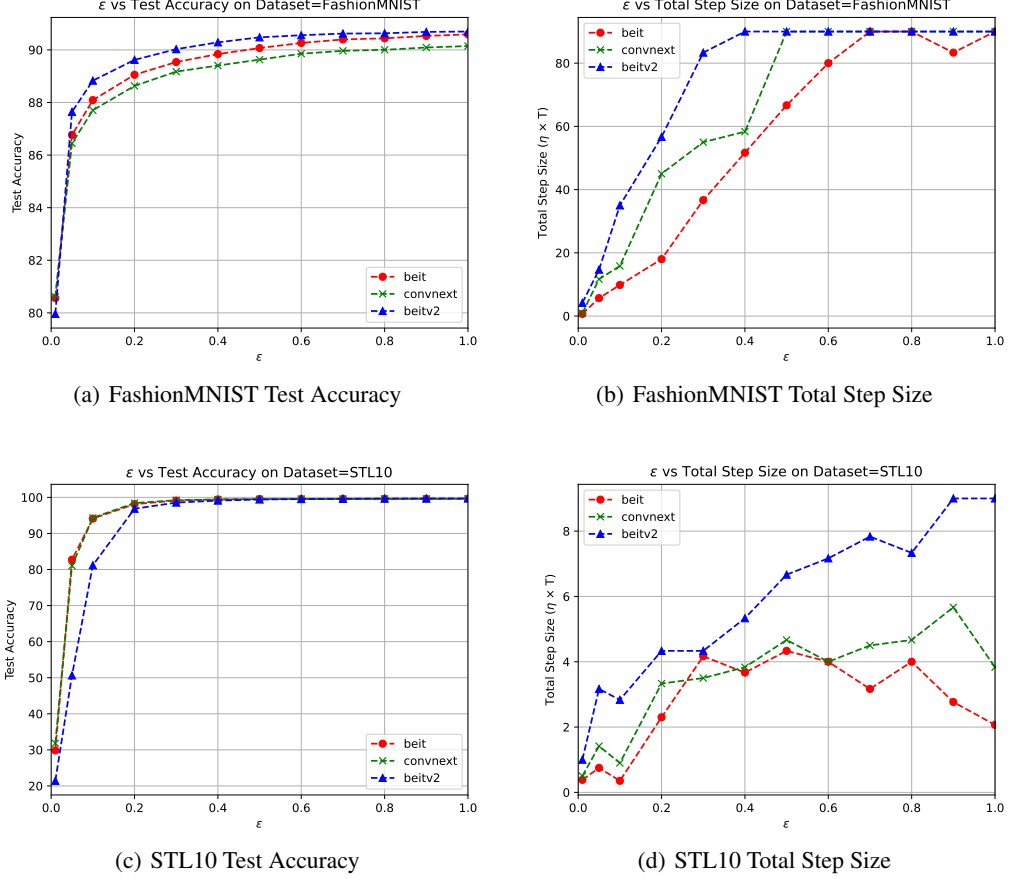

(a) FashionMNIST Test Accuracy

(b) FashionMNIST Total Step Size

(c) STL10 Test Accuracy

(d) STL10 Total Step Size

Figure 22: Pareto frontier for $\varepsilon$ vs test accuracy and total step size for STL10 and FashionMNIST. Beitv2 excels for larger values of $\varepsilon$ but beit and convnext are better for smaller values of $\varepsilon$. The inflection point varies across datasets.