# OpenReview forum: "A New Linear Scaling Rule for Differentially Private Hyperparameter Optimization"
_NeurIPS.cc/2023/Conference — Submitted to NeurIPS 2023_

### Official Review · Reviewer_KKeT · 2023-07-03

**Soundness:** 1 poor
**Presentation:** 3 good
**Contribution:** 3 good
**Rating:** 3
**Confidence:** 4

**Summary:**

This paper presents a novel hyperparameter tuning method in the presence of a privacy budget: linearly extrapolating from observations with very low privacy loss.

**Strengths:**

The core technique presented here is certainly interesting and deserving of future study. The paper tackles an issue which is often unaddressed in the literature on training DP models: that of choosing hyperparameters subject to a privacy budget. This problem itself is also deserving of further study.

**Weaknesses:**

* A primarily empirical paper will live and die with the strength of its baselines (as well as its upper bounds in a case like this one where upper bounds on the efficacy of the technique can be computed). The baselines here are insufficiently strong, and do not seem to reflect the statements in the cited papers. The core technique _could_ be a component of a strong paper, but this paper is not it.

* Some baseline issues: the citation problems with [51], [52] (detailed below). Lack of comparison to the 'naive baseline' of directly applying gaussian mechanism to results of grid search, say given known training statistics / optimal hparam values for nonprivate datasets (to avoid infinite regress, and here not so much of a problem since the experiments are all focused on public feature extractor settings). Lack of clear comparison to the 'upper bound' of _forgetting_ about the privacy cost of hparam search, which _should_ be an upper bound in _all_ scenarios considered here (IE, performing a sufficiently large grid search directly targeted at the problem at hand).

* On [51]/[52], I see the reporeted CIFAR10 numbers from [51] as 98.8\% at $\epsilon=1$ and 98.9 at $\epsilon=\infty$ (table 1 of [the arxiv version](https://arxiv.org/pdf/2211.13403.pdf)). Is there a typo in figure 2 of the paper under submission? Similarly, [51] seems to claim 88.1\% and 90.6\% at the $\epsilon=1, \infty$ for CIFAR-100. I uncovered these discrepancies since the paper under submission seemed to present implausibly strong results to me--e.g. it should be _impossible_ to achieve at epsilon=1 what none of the cited papers achieved at epsilon=\infty just by tuning hyperparameters (see figure 2).

* The statements of timing on Imagenet seem wrong? The cited paper [51] seems to be pointing to a version from Nov 2022, clicking through to [52] seems to show a version uploaded in May 2022--so where are Jan 2023 and 'within the last month' coming from?

* Some more consideration required in decomposition of $r$--do we know that random decomposition 'is enough'? Presumably it's not, since we _can_ generate an $\eta$ for which the problem will presumably diverge?

**Questions:**

The major questions I have for the authors here are the sources for both the claims on timing in relation to ImageNet (see weaknesses above) and the issues in citations (particularly with [51] and [52]).

**Limitations:**

Societal impact not immediately applicable.

---

> ### Author Rebuttal · Authors · 2023-08-02
>
> > The core technique presented here is certainly interesting and deserving of future study.
>
> We appreciate the reviewer's recognition that we have chosen to study an important problem. The majority of our rebuttal will engage with the main stated weakness, that we do not compare with the right numbers in the cited works or consider the right baselines.
>
> > A primarily empirical paper will live and die with the strength of its baselines (as well as its upper bounds in a case like this one where upper bounds on the efficacy of the technique can be computed). The baselines here are insufficiently strong, and do not seem to reflect the statements in the cited papers.
>
> We now explain that we already compare to the baselines recommended by the reviewer, and compare to the right numbers in the cited works. As a reminder, we provide Thm 2.3 that holds for the linear probing we consider for CV tasks.
>
> > Lack of comparison to the 'naive baseline' ... Lack of clear comparison to the 'upper bound' of forgetting about the privacy cost of hparam search ...
>
> The 'upper bound' grid search that does not consider the privacy cost of hparam search is the exact 'upper bound' considered in Figure 3; we compare the performance of the linear scaling rule to the 'upper bound' grid search. That is, for the x-axis point labeled 1.0 (eps=1.0), we evaluate 100 hyperparameter combinations for grid search with eps=1.0 (see Fig 19/20) and do not consider the privacy cost of any of them but the optimal hyperparameter combination. As noted in line 184, we do not compare to the 'naive baseline' because it will distort the x-axis too much, and we feel the linear scaling rule is sufficiently competitive with the 'upper bound'. If we were to chart the 'naive baseline' then the x-axis would stretch to eps=10 since the privacy cost would grow with $\sqrt{trials}$.
>
> > On [51]/[52], I see the reported CIFAR10 numbers from [51] as 98.8% at eps=1
>  and 98.9 at eps=∞ (table 1 of the arxiv version).
>
> The referenced numbers are from models pretrained on JFT, a massive proprietary Google dataset that we cannot access (note the rightmost column of Table 1 in [51]). In the caption of Figure 2 we specify that we are considered models pretrained on ImageNet-21k, and therefore use the numbers from the middle row in Table 3/Table 4.
>
> > Is there a typo in figure 2 of the paper under submission?
>
> We don't believe there is a typo here.
>
> > Similarly, [51] seems to claim 88.1% and 90.6% at the eps=1,∞ for CIFAR-100.
>
> Similarly, these numbers are for pretraining on JFT, which is an unfair comparison since JFT is much much much larger (3 billion images) than ImageNet-21k.
>
> > I uncovered these discrepancies since the paper under submission seemed to present implausibly strong results to me...
>
> We provide all the code necessary to reproduce our results in the first line of the Appendix (also made available to the AC) and we urge the reviewer to choose any of the models in Figure 5 (see utils.py), extract the features, and do linear probing with the given hyperparameters; it will only take a few minutes. The improvements over the results reported in cited papers stem using better models and more optimal hyperparameters. For example if we use the same beit model as [7] we can in fact improve over their best results as shown in Figure 5. It's entirely possible that the papers did not consider the best hyperparameters, because the search space is vast (learning rate * schedule * epochs * batch size * momentum * optimizer * parameters to optimize * initialization).
>
> > The statements of timing on Imagenet seem wrong? ...
>
> We are quoting the timing and numbers from the published versions not the Arxiv versions, but we cited the Arxiv version in the bibtex; thank you for bringing this error in the bibtex to our attention, we will make sure to fix it. The published versions are public on OpenReview;  [51] https://openreview.net/forum?id=Uu8WwCFpQv [52] https://openreview.net/forum?id=Cj6pLclmwT.
> We provided these times in the caption of Figure 4 in order to provide a rough timeline on 'SOTA' for any future readers, because we assume that 'SOTA' will advance quickly. We never claim in the main paper that the gap between our method and [51] is due to the recency of the work, but rather because they pretrain on a much better dataset than anything we have access to. Furthermore, we provide results on full fine-tuning language tasks in addition to ImageNet, whereas their proposed method (DP-FC, [51]) explicitly only works for linear probing on vision tasks.
>
> > Some more consideration required in decomposition of r...
>
> Based on the heatmaps in the Appendix, specifically Figure 19, we can see that the decomposition does not hold for r=eta (single-step), but this is not a critical issue because [52] uses single-step exclusively, it is just suboptimal. It's true that if r=1000 we definitely don't want to use eta=1000, but as we note in line 107 we are not really interested in the behavior with very large values of epsilon. The exact routine we used in decomposing r is a simple nested for loop, that iterates over randomly shuffled arrays of valid epoch and learning rate values and checks whether their product is within some tolerance of the given r.
>
> > The major questions I have for the authors...
>
> We hope that the clarification on the publication date of [51, 52] at TMLR rather than the Arxiv date can address the first question (we only put these times in the caption of Figure 4 in order to provide a rough timeline on 'SOTA' for any future readers, because we assume that 'SOTA' will advance quickly). For the second question, we hope that the clarification that we are comparing our ImageNet-21k results to the ImageNet-21k results in [51, 52] explains the numbers in Figure 2. We believe that comparing our ImageNet-21k results to their JFT results would be unfair because we do not have the ability to evaluate on JFT, and nor do any researchers outside of Google.

---

> > ### Comment · Reviewer_KKeT · 2023-08-13
> >
> > I appreciate the authors' high-spirited response. However, I believe the issues with figure 2 remain too strong to alter my score.
> >
> > Fundamentally, the reason that I am unwilling to move is the fact that $\epsilon=\infty$ should _always be able to beat, or at least match, any DP hyperparameter tuning method_. Therefore, given that the authors seem to have reproduced these results from scratch rather than quoting them, many questions remain on the methods of generating the numbers in figure 2.

---

> > > ### Author Response · Authors · 2023-08-13
> > >
> > > We appreciate the reviewers' diligent efforts in providing insightful comments that are helpful in enhancing the quality of our paper. In the forthcoming revisions, we will craft a more persuasive discourse to assure that our results are valid and not 'too good to be true'. To summarize, the improvements are largely attributed to enhancements in the pretrained models, which can achieve impressive performance even in zero-shot scenarios.
> > >
> > > All numbers in Figure 2 are quoted from the papers where they are drawn. For [51], the numbers are drawn from their Table 3, second multirow (ImageNet-21k), column 5 ($\varepsilon=1.0$), row 5 (their best result with their DP-FC method) which is 96.3, and for the $\varepsilon=\infty$ it is from the column titled 'non-private' in that same multirow (96.6).
> > >
> > > We obtain better results even compared to the $\varepsilon=\infty$ numbers in other papers via a combination of better models and better hyperparameters. We compare a range of models in Figure 5. Some of these perform better than others, and prior work also uses different models for linear probing. For example, [7] uses beit, [51] uses vit-L, and [15] use WRN-40-4. As we note in line 252, the best model for DP fine-tuning is also the model with the highest non-private accuracy as reported by [78].
> > >
> > > Furthermore, we have provided the code necessary to reproduce our results in a matter of minutes (shared with the AC); even running the code for 10 seconds should be sufficient to verify our claims.

---

> > > ### Author Response · Authors · 2023-08-21
> > >
> > > As the discussion period is drawing to a close, we would like to thank you for your efforts in reviewing the paper. As shown by the extensive discussion with other reviewers, which resulted in an overall score increase of 3 points across the other reviews, we were able to clarify that the improvements shown by our method over prior work is largely due to the use of better pretrained models. When we compare to a work that uses the same pretrained model that we do such as [7], the improvement is due to better hyperparameters as found by our method and the tricks we propose in Appendix A.2. There are other (unpublished) papers that also improve over the same pretrained model in [7] as we discussed with reviewer kJWb and we will add more entries to our Table 2 to compare with those.
> > >
> > > In your initial review you mentioned a lack of comparison to other baselines. We hope our initial response clarified that we did compare to the upper bound (non private grid search) in the main paper. We also want to draw your attention to the discussion on the main AC comment thread, the response to reviewer kJWb, and the response to reviewer BCCV, where we do fair comparisons to three other baselines: random search, the method in [3], and the recent/concurrent method in [4]. We improve upon all these by multiple percentage points. For example, the error rate reduction between random search and our method as compared to the oracle is about 70% (exact numbers are in that other comment). We will include all these additional baseline comparisons in the camera ready.
> > >
> > > We will incorporate the extensive discussion with other reviewers into the camera ready to ensure that our improvements do not come across as 'too good to be true'. We would be grateful if you can leave a final response based on these discussions.

---

### Official Review · Reviewer_kJWb · 2023-07-07

**Soundness:** 2 fair
**Presentation:** 2 fair
**Contribution:** 2 fair
**Rating:** 5
**Confidence:** 3

**Summary:**

The paper proposes a linear scaling rule for finding the optimal value of the learning rate and number of training steps for differentially private SGD (DP-SGD). The idea is simple, small amount of privacy budgets are allocated for two initial DP learning rate optimization procedures, and then the values are extrapolated to bigger epsilon-values using linear scaling (as a function of epsilon). The work is mostly experimental, and the experimental results e.g. with CIFAR-10 show that for epsilon between 0 and 1, the scaling rule seems to nicely fit the optimal values found by the grid search.

**Strengths:**

- The idea seems very interesting and novel.
- The paper is mostly written well and is easily readable.


**Weaknesses:**

- The technique is restricted to optimising the learning rate and the length of the training. I wonder if similar extrapolation (perhaps more generally polynomial extrapolation) could be used to find optimal the optimal hyperparameter values for other hyperparameters.

- The technical part could be written more carefully. It remains unclear whether you use RDP or GDP. The hyperparameter tuning cost of the method by Papernot and Steinke is in terms of RDP, but you list theoretical results in terms of GDP. In the end of Alg. 1 you write that the total cost is "$\varepsilon_f + \varepsilon_0 + \varepsilon_1$". Is that approximate DP? In case you use the classical composition result where you just add up the privacy parameters, what happens to the $\delta$-parameters?

- Some conclusions are a vaguely formulated/confusing. On p. 7 you have the subtitle "Linear Scaling is robust to distribution shifts", but then you seem to show and also claim in the subsequent text that DP itself is robust to distribution shifts. Somehow the message is vague here.

- The contribution remains too thin in my opinion. There is really no theoretical or even heuristic explanation for the proposed scaling rule. There two theoretical results given, a GDP composition result (which is well known and should be cited as such) and another result of which importance I find difficult to judge.


**Questions:**



What does the following line in Alg. 1 mean: "Use privacy loss variable accounting to calibrate noise parameter $\sigma$ given $\varepsilon$" ?

- I somehow find it hard to believe that you would get 99 percent test accuracy for CIFAR-10 with $\varepsilon=1.0$. The SOTA results by De at al. ("Unlocking High-Accuracy Differentially Private Image Classification through Scale") are somewhere at 95 percent for  $\varepsilon=1.0$. What made you to get this good results?

- You mention in the description of the method that you scale up to $\varepsilon=1.0$. Why is that? You seem to use varying $\varepsilon$-values in the experiments.

There seem to be typos here and there, please go through the writing carefully. Here few examples:
line 387-388: " treat < 10% of the private training dataset and public" -> "as public"
line 394: "We provide find that our method attains..."

I don't quite understand the following sentence:
"The key assumption in DP fine-tuning is that there is no privacy leakage between public data and private data."
You mean that there are not too big distribution shifts?

**Limitations:**

Some of the limitations are discussed in Section 5 but it could be expanded I think.

---

> ### Author Rebuttal · Authors · 2023-08-02
>
> We thank the reviewer. We address concerns about scope and clarity of contributions, and explain how we improve over the prior SOTA.
>
> >The technique
>
> There are a number of hyperparameters for DP; clipping norm, batch size, momentum, optimizer, what parameters to update and how to initialize them. We design our linear scaling rule around analytical optimal choices for these hyperparameters. These choices are mentioned in line 94 and 95 in the main body and detailed ablations are in Appendix A.2 (Table 1). We do not need to estimate the clipping norm because we know that unit clipping norm is optimal, same for using full batch updates, etc. The remaining hyperparameters we are left with are just the learning rate and number of iterations, and our method provides guidance on optimizing these.
>
> >The technical
>
> We appreciate the reviewer's careful reading of our theory in Section 2. We have incorporated these comments into the following overhaul. First we will rewrite Lines 125-134 and update the corresponding lines in Alg. 1 with the following procedure. Given a desired final $(\epsilon, \delta)$-guarantee, we will use the GDP-approx DP conversion from Corollary 2.13 in [18] to find the appropriate value of the parameter $\mu$ for GDP. Then we will allocate $\mu$ across the hyperparameter optimization runs and final run according to $\mu = \sqrt{3 \mu_{1}^{2} + 3 \mu_{2}^{2} + \mu_{f}^{2}}$, where $\mu_{f}$ is the privacy parameter for the final run that uses the hyperparameters adaptively chosen by using the linear scaling rule on the outputs of the runs with smaller privacy budgets $\mu_{1}, \mu_{2}$. Then for a given $\mu_{i}$, we will decompose this into $(T, \sigma)$ according to Proposition 2.1 in the main paper.
> By writing the privacy analysis in this manner we can stay in GDP the entire time, and we hope these presentation changes will address the reviewer's feedback.
>
> We are only using GDP for our method, and we do not use the RDP method from Papernot and Steinke. Their method requires doing a random search and cannot incorporate priors (i.e., is not adaptive like ours). The point of our work is to develop a principled heuristic, that works in practice -across 20 datasets, CV and NLP, linear models and full finetuning of transformers- such that we don't need to pay much privacy cost for tuning. Fixing a random number of runs as they do will not provide a good approximation of the optimal hyperparameters unless we pay a large privacy cost.
>
> >The contribution
>
> We summarize the contribution of our work in terms of experiments, theory and heuristic analysis.
>
> As an empirical work, we provide new open-source SOTA baselines for DP research across 20 tasks that are efficient to reproduce and easy to build on top of by just downloading the pretrained models and extracting features. Our hyperparameter optimization method removes the computational burden of optimizing hyperparameters anew for DP training. We believe our work has the potential to help the community build research on the cutting edge by providing efficient and reproducible baselines.
>
> We provide Thm 2.3 (that holds for the linear probing we do for CV tasks) as a theoretical explanation of our scaling rule and validate it empirically (Fig3/Fig4). Thm 2.3 states that when the noise is not too large and the learning rate is smaller than some data-dependent quantity, DP and non-DP linear probing converge to the same solution. The noise increases with the number of iterations T, so we scale up T and the learning rate with $\varepsilon$ while adaptively estimating them to get a good data-dependent approximation of the optimal learning rate.
>
> We provide a heuristic explanation in line 108. Here is another: In order to satisfy indistinguishability over a larger set of final models, we must increase $\varepsilon$. The size of this set increases with the product of the learning rate and number of iterations. So we should scale these with $\varepsilon$.
>
> As mentioned previously we will rewrite the composition analysis to properly cite the GDP composition result (the current citation is in Appendix A.5).
>
> >I somehow
>
> We provided everything needed to reproduce our experiments (including open source code). We encourage the reviewer to run our experiments. For CIFAR10 $\varepsilon=1.0$ 99% test accuracy comes from the combination of a better pretrained model than De et al. (see the models we evaluate in Figure 5) and better choices of the optimal hyperparameters. Other papers (Table 2) have also improved on De et al. One of these papers, Bu et al. ([7] in the main paper) uses the same pretrained model that we do (beit) but we obtain better results because we use better hyperparameters (see Table 1 in the Appendix, A.2).
>
> >I don't
>
> If we assume the existence of some publicly available data for pretraining and then do DP fine-tuning on the private data, it's crucial that there is no privacy leakage between the public data and private data. There is only 0 distribution shift when public = private, and this violates the key assumption (no privacy leakage because public and private data are sufficiently different) in DP fine-tuning. If the public data is so different from the private data that it can be used for pretraining without privacy leakage, there must be some distribution shift. This motivates our analysis of the robustness of DP models trained with the linear scaling rule to distribution shifts in Page 7.
>
> >You mention
>
> We write "eps << 1 and scaling these up to eps = 1" just to concretely fix eps, rather than writing something like "eps << eps* and scaling these up to eps = eps*" which might be confusing.
>
> >Some conclusions
>
> The distinction is that models trained with DP typically have less accuracy and may be vacuously more robust (there is no gap between ID and OOD when both have 0% accuracy), but we find that we can get robustness without sacrificing much accuracy. We will clarify this.
>
> >Typos
>
> We will fix these typos.

---

> > ### Comment · Reviewer_kJWb · 2023-08-16
> >
> > Thank you for the replies! That GDP accounting formula looks correct and should be used in the revised paper. I have continued discussion in the other thread.

---

> > > ### Comment · Reviewer_kJWb · 2023-08-16
> > >
> > > I was reading the rebuttal to reviewer BCCV. When you compare to the mentioned papers [3] and [4], don't you have a different setting in a sense that you have full batch training whereas they have mini-batch algorithms?  I mean it does not sounds like an apples to apples comparison. I would imagine that comparing to those methods here appropriately (full-batch training, optimise only the learning rate) you might get similar results.

---

> > > > ### Author Response · Authors · 2023-08-16
> > > > **Clarification of Rebuttal to Reviewer BCCV**
> > > >
> > > > Thank you for your comment. First we note that our proposed method tunes both the learning rate and number of iterations. [3] tunes only the learning rate, so we evaluate it over the grid of epoch values and report the best results. Next we give detailed analyses of the apples to apples comparisons to [3] and [4].
> > > >
> > > > Our result for DPAdamWOSM ([3]) is a reimplementation of that paper, as they don’t report results with the settings we consider. The code can be found in `wosm_impl.py`. There, we are keeping all the other hyperparameters the same such as batch size except for what DPAdamWOSM tunes, which is the learning rate. Therefore, it is an apples to apples comparison.
> > > >
> > > > As for the comparison to [4], the authors note that “3.4 Computational Savings Our scaling approach for DP-SGD described in Section 3.2 implies that the subsampling ratio γ, the noise level σ and the number of iterations T are the same when evaluating the private candidate models using the tuning set X1 and when evaluating the final model using the larger dataset.” That is, they state here explicitly (and also mention in the previous section) that their method requires holding the subsampling rate constant between the candidate models that are evaluated on small datasets and the final model with the larger dataset. Our approach uses subsampling rate=1 (that is, we use the entire dataset rather than using a subset of the dataset as they do); this is considered in their Figure 1, where it seems that using full-batch training either makes their method worse than the baseline (blue dotted line, variant 2, that obtains better results in Figure 3 where they are tuning for the learning rate and number of iterations as we are) or the same as the baseline (green dotted line, variant 1). From our analysis of this figure, it seems that comparing their method to our setting of full-batch training would only degrade their performance.
> > > >
> > > > Doing an apples to apples comparison between [4] and our method would therefore entail modifying their approach to use $\gamma=1$ and rerunning their experimental protocol. Given that [4] is a somewhat recent work that does not appear to provide code, this is challenging. We might instead look at their results from Fig. 5 where they tune the same hyperparameters that we do (learning rate and number of iterations) and apply the original linear scaling rule (https://arxiv.org/pdf/1706.02677.pdf) to scale up their optimal learning rate for $\varepsilon=1$ from $\eta=0.15$ to $\eta = 0.15 \times \frac{1}{\gamma} = 0.15 \times \frac{1}{0.02} = 7.5$. This does not perform well because the learning rate is too large; we could instead just use their original learning rate of $\eta=0.15$ but this does not perform well because the learning rate is too small.

---

> > > > > ### Comment · Reviewer_kJWb · 2023-08-16
> > > > >
> > > > > > Doing an apples to apples comparison between [4] and our method would therefore entail modifying their approach to use $\gamma=1$ and rerunning their experimental protocol.
> > > > >
> > > > > I also think that would be required, without an experimental comparison it is difficult to say. Considering they scale the learning rate as well, for these learning rate values you use in your CIFAR10 and CIFAR100 experiments, and using the models you are using, I would expect the final model hyperparameter values to end up somewhere to the right end of the grid and thus give similar results.

---

> > > > > > ### Author Response · Authors · 2023-08-16
> > > > > >
> > > > > > > I also think that would be required, without an experimental comparison it is difficult to say.
> > > > > >
> > > > > > Thank you for your comment; we have reached out to the authors of [4] to request the code for their paper so that we may do the experimental comparison. We hope they can respond, but they may also be tied up with rebuttals, so we just want to temper the reviewer's expectations. At this time we do not anticipate being able to implement their paper from scratch and run the necessary experiments by the rebuttal deadline.
> > > > > >
> > > > > > We want to reiterate that even if their final model hyperparameter values were to end up somewhere in the correct region, using a subsampling rate of 1 would likely defeat the point of their paper per Figure 1, as their method controls the privacy cost of hyperparameter candidates via using small batches as you have already noted. This reduces their method to the baseline, which does the hyperparameter search without small batches, and the performance of that baseline is already reported throughout the paper as being significantly worse than their method. That is, in order to do the apples to apples comparison we would have to remove subsampling and this would increase the privacy cost of their final models to no longer be $\varepsilon=1$, but instead be multiple times larger.

---

> > > > > > > ### Comment · Reviewer_kJWb · 2023-08-16
> > > > > > >
> > > > > > > Looking at the mentioned [4], here might be a misunderstanding, I don't think it would match with that baseline.

---

> > > > > > > > ### Author Response · Authors · 2023-08-16
> > > > > > > >
> > > > > > > > Perhaps you can clarify what the desired experimental evaluation is? Your comment was;
> > > > > > > >
> > > > > > > > > comparisons to [4] do not seem necessarily fair since they have small batches as hyperparameter candidates
> > > > > > > >
> > > > > > > > We are saying that the way to evaluate their method, which does the baseline algorithm for a subsampled dataset for each hyperparameter candidate in order to limit the privacy cost of evaluating that hyperparameter candidate, in our full-batch GD setting, would be to instead run the baseline algorithm with the full dataset. This corresponds in Figure 1 to setting the subsampling rate to 1.0. This seems that it would match that baseline.

---

> > > > > > > > > ### Comment · Reviewer_kJWb · 2023-08-16
> > > > > > > > >
> > > > > > > > > In their Fig. 1 the subsampling rate is the one used by the tuning algorithm, whereas I meant that one could use subsampling ratio 1 for the underlying DP-SGD that is being tuned. It would still give the amplification for the tuning algorithm,
> > > > > > > > >
> > > > > > > > > I am not expecting authors to make any additional experiments at this point, however I believe one could end up to similar results with other tuning algorithms. Also the method by Papernot and Steinke (2022) would be one baseline to consider: their method commonly doubles or triples the epsilons while allowing much many more candidates to be evaluated. So if you would choose your learning rate grid blindly e.g. from 1e-5 to 1e2 and have e.g. 30 different values, which might require more evaluations using your method, I think one could come up with settings where the baseline would outshine. Especially considering that having $\epsilon/3$ instead of $\epsilon$ for the DP cost does not seem that crucial for these particular final layer training / fine-tuning tasks.
> > > > > > > > >
> > > > > > > > > These discussions have clarified a lot and I have quite a different opinion about the paper I initially had! I am raising my score, however I suspect the paper would need more work / polishing / clearer focus to reach the bar.

---

> > > > > > > > > > ### Author Response · Authors · 2023-08-16
> > > > > > > > > >
> > > > > > > > > > Thank you for the clarification. This indeed sounds like an interesting experiment to run and we will add it to our queue.
> > > > > > > > > >
> > > > > > > > > > We appreciate your willingness to engage in discussion and are thankful for your increased score. We understand that the key points of our paper should be readily understood without so much back-and-forth discussion. We will deliver a version of the paper that more clearly conveys our main contributions, methods, and findings for the camera ready.

---

> > > > > > > > > > ### Author Response · Authors · 2023-08-21
> > > > > > > > > >
> > > > > > > > > > We ran the requested experiment on CIFAR100 with the requested range of learning rates between 1e-5 to 1e2 by changing the learning rate grid to `valid_etas = np.power(10, np.arange(-5, 2, 0.15))`. This range is quite large (indeed, it's larger than the range in [4]) so we hope this satisfies your comments on both grid granularity and scale. Here are the results for random search over 10 trials:
> > > > > > > > > >
> > > > > > > > > > ```
> > > > > > > > > > All searched etas: 0.00251, 0.00008, 1.25893, 56.23413, 0.31623, 0.01000, 0.00006, 0.00355, 0.31623, 19.95262
> > > > > > > > > > All searched ts: [130, 25, 120, 120, 5, 85, 85, 15, 60, 65]
> > > > > > > > > > All searched sigmas: 42.56, 18.66, 40.89, 40.89, 8.35, 34.42, 34.42, 14.46, 28.91, 30.10
> > > > > > > > > > All searched accs: [84.76, 84.12, 89.57, 75.74, 83.72, 84.89, 83.62, 83.39, 88.07, 86.56]
> > > > > > > > > > Mean and standard deviation of accs: 84.44 +- 3.50
> > > > > > > > > > ```
> > > > > > > > > >
> > > > > > > > > > Note that as you predicted making a logarithmic grid will make the baseline worse. The average accuracy is not great and the standard deviation is high, but nearly all the results are still $>80$% accuracy because the pretrained model is good. We do not view the strength of the pretrained models as a shortcoming of our method, because as we will see, the standard by which to evaluate our method is how much it improves over the baselines as compared to the upper bound/oracle (grid search where we don't account for the privacy cost of hyperparameter tuning).
> > > > > > > > > >
> > > > > > > > > > And here is the result for our method (we only had time for one run). We used $\varepsilon_1=0.01, \varepsilon_2=0.05$ and did $n=47$ runs with $\varepsilon_1$ and $n=11$ runs with $\varepsilon_2$. The output of `prv_accountant` here is `Estimates of epsilon: lower bound 0.99981 estimate 0.99982 upper bound 0.99983`, so all we changed was improving our utilization of the total privacy budget $\varepsilon=1$. **We still get $89$% accuracy, improving over random search by $4$%. The relative error rate reduction is 84.54%**
> > > > > > > > > >
> > > > > > > > > > ```
> > > > > > > > > > Sorted configs for eps_1 [{'lr': 0.0012589254117941816, 't': 1, 'sigma': 255.7393275537945, 'acc': 3.44} (output truncated for readability)
> > > > > > > > > > Sorted configs for eps_2 [{'lr': 0.22387211385683928, 't': 55, 'sigma': 434.19948928919797, 'acc': 27.4}] (output truncated for readability)
> > > > > > > > > > Doing linear interpolation with eps [0.01, 0.05], rs [0.0012589254117941816, 12.31296626212616]
> > > > > > > > > > Extrapolated value of r: 295.48
> > > > > > > > > > Launching run with lr 3.54813, t 85, sigma 35.38
> > > > > > > > > > Final accuracy: 88.98
> > > > > > > > > > All searched etas (excluding eps_1 for readability): 5.01187, 0.11220, 0.00126, 1.25893, 0.00126, 56.23413, 56.23413, 5.01187, 0.22387, 56.23413, 0.63096, 3.54813
> > > > > > > > > > All searched ts (excluding eps_1 for readability): [15, 25, 50, 50, 140, 100, 85, 10, 55, 50, 55, 85]
> > > > > > > > > > All searched sigmas (excluding eps_1 for readability): 226.38, 292.58, 413.77, 413.77, 691.86, 585.16, 540.13, 185.15, 434.20, 413.77, 434.20, 35.38
> > > > > > > > > > All searched accs (excluding eps_1 for readability): [23.41, 19.92, 11.61, 18.85, 13.85, 11.13, 12.16, 25.25, 27.4, 12.88, 25.69, 88.98]
> > > > > > > > > > Total privacy cost for final accuracy including the privacy cost of hyperparameter search 0.99982
> > > > > > > > > > ```
> > > > > > > > > > We can estimate the variance by doing 3 random decompositions over r. This gives us;
> > > > > > > > > > ```
> > > > > > > > > > Launching run with lr 14.12538, t 20, sigma 17.16
> > > > > > > > > > Final accuracy: 89.5
> > > > > > > > > > Launching run with lr 2.51189, t 120, sigma 42.04
> > > > > > > > > > Final accuracy: 88.83
> > > > > > > > > > ```
> > > > > > > > > > Analyzed this way, the mean and standard deviation over 3 runs are $89.10$% with standard deviation $0.287$. Further ways to estimate the variance would be to do multiple runs with those decomposed values. We would also run the method multiple times.
> > > > > > > > > >
> > > > > > > > > > Note that for computational efficiency we just made the smallest privacy budget trials all use the same number of epochs so that we could precompute all the noise multipliers, because running `prv_accountant` for $\varepsilon=0.01$ was taking a long time.
> > > > > > > > > >
> > > > > > > > > > Our method improves over random search in 2 regards.
> > > > > > > > > > First, recall that our method intelligently allocates privacy budget to different phases of hyperparameter tuning. We can test a lot of values very cheaply for small privacy cost in the first round of tuning. Even though as you noted we require more evaluations to hit a good value, we can just evaluate the entire learning rate grid.
> > > > > > > > > > Second recall that our method incorporates the intuitive prior of linear scaling. The second round of tuning will only consider values larger than the best value from the first round. So even though the grid is large and most of the values are bad, the first round can narrow down the grid significantly. This still isn't perfect (in the above trace, we evaluated the massive and obviously incorrect learning rate of $56.23$ three times!) but it helps a lot.
> > > > > > > > > >
> > > > > > > > > > Thanks again for your review! We don't expect you to change your score, we genuinely just thought your proposed experiment was a good idea and wanted to report the results. We still planning on doing a comparison to the method of Papernot and Steinke and [4] for the camera ready, but those will take a bit more time and the discussion period is over.

---

### Official Review · Reviewer_BCCV · 2023-07-08

**Soundness:** 3 good
**Presentation:** 3 good
**Contribution:** 4 excellent
**Rating:** 6
**Confidence:** 4

**Summary:**

This study proposes a new algorithm for privately selecting hyperparameters subject to maximizing the model utility. The new algorithm draws inspiration from the linear scaling rule that suggests increasing learning rate as batch size increases. Given the number of hyperparameters in DP-SGD the proposed algorithm simply scales learning rate and number of iterations as the privacy budget increases. This introduces a new hyperparameter that is selected privately with a portion of the privacy budget while the rest is used to perform the normal hyperparameter search. The study provides brief theoretical intuition for why we can expect this linear scaling rule to more efficiently determine optimal hyperparamters compared to previous methods and extensive empirical evidence on 20 different benchmark datasets.

**Strengths:**

- One of the first papers to demonstrate improved privacy-utility tradeoffs that takes hyperparameter tuning into account. This is substantial as the field has mainly focused on evaluating the privacy-uility tradeoff without considering the privacy cost of hyperparameter tuning. As we move towards more practical implementations, this will be necessary.
- Clever use of the linear scaling rule to perform hyperparameter search and the resulting algorithm is simple to use.
- Extensive empirical evaluation and insightful analysis. For example, very few analyses have been done on the intersection of DP and distriutional shift. Yet, this linear scaling rule that is proposed holds in the presence of distribution shift.


**Weaknesses:**

- “We are 165 the first to show that DP-SGD is capable of learning to handle distribution shifts without using any 166 techniques from the distributionally robust optimization (DRO) literature” -> There are a couple of other papers that draw this connection. [1,2]
- Lack of comparison to other private hyperparameter selection algorithms or hyperparameter free private learning algorithms [3, 4]
- Unclear why the initial hyperparameter search can be done with such a small privacy budget even though this is a key factor driving the performance of the algorithm.

[1] Kulynych, Bogdan, et al. "What you see is what you get: Distributional generalization for algorithm design in deep learning." arXiv preprint arXiv:2204.03230 (2022): 13.
[2] Hulkund, Neha, et al. "Limits of Algorithmic Stability for Distributional Generalization." (2022).
[3] Mohapatra, Shubhankar, et al. "The role of adaptive optimizers for honest private hyperparameter selection." Proceedings of the aaai conference on artificial intelligence. Vol. 36. No. 7. 2022
[4] Koskela, Antti, and Tejas Kulkarni. "Practical differentially private hyperparameter tuning with subsampling." arXiv preprint arXiv:2301.11989 (2023).


**Questions:**

1. What is the intuition for why the initial hyperparameter search can be done with such a small privacy budget? Is it possible to simply randomly initialize $r_0$ and achieve similar performance?
2. How does this method compare with optimization algorithms that reduce the need for hyperparameter tuning?


**Limitations:**

The paper does address the technical limitations of the paper (specifically the assumption of access to public and private data). The main improvement for the limitations is to address the comparison to other tuning algorithms or optimization algorithms that don’t require as much tuning.

---

> ### Author Rebuttal · Authors · 2023-08-02
>
> > The main improvement for the limitations is to address the comparison to other tuning algorithms or optimization algorithms that don’t require as much tuning.
>
> We appreciate the reviewer's feedback and care in bringing these papers to our attention. In this rebuttal we provide a detailed comparison to the papers you cited on DRO [1,2] and find that our method outperforms the other tuning alternatives in [3, 4].
>
> > We are the first to show...
>
> We appreciate you bringing these papers to our attention. We will update the contribution to read "We show that DP-SGD provides robustness to covariate, subpopulation and label distribution shifts for synthetic and natural datasets." We will also provide the following comparison of the two papers you cited.
> [1] proposes DP-IS-SGD that improves the robustness of DP-SGD by removing per-sample gradient clipping (therefore removing the introduced bias but also losing the privacy guarantee; see 4.2 in [1]) and uses knowledge of the groups to sample subpopulations at different rates to improve robustness. Because our method uses DP-GD to maximize the signal-to-noise ratio of updates (Appendix A.2) and requires clipping (because our primary goal is the privacy guarantee, unlike [1] which focuses on DRO) and we do not assume knowledge of groups, we cannot make use of DP-IS-SGD.
> [2] seems to be a recent work (Google shows only a rejected ICLR2023 submission) and they conclude that "[DP-SGD] is not a good candidate for improving robustness under covariate or subpopulation shift, as it comes
> at a major cost to accuracy." (page 7) This conclusion runs counter to our findings, and we believe the reason is because the numerical findings from [2] are not conclusive. The error bars are very large and the results are somewhat conflicting with each other. Our interpretation of their results is that because their DP-SGD degrades accuracy, it should also increase robustness; however we find that even when DP-SGD does not degrade accuracy it still improves robustness (Figure 6 in our main paper).
>
> > Lack of comparison to other private hyperparameter selection algorithms...
>
> We appreciate the reviewer drawing our attention to these algorithms. We now provide a comparison. At a high level we evaluate both DPAdamWOSM [3] and the RDP tuning [4] and find that our linear scaling rule outperforms these methods, and analyze why.
>
> We implement DPAdamWOSM [3] and tune the necessary hyperparameter T (# of epochs) between 1 and 200 and report the performance for the best value of T for a fixed $\varepsilon=1$ without accounting for the privacy cost of this tuning. On ImageNet DPAdamWOSM achieves 79% at $T=50$, which is 4% lower than our method (83% at $\eta=20, T=50$). At a high level, our linear scaling rule attempts to do a data-dependent learning rate selection, while DPAdamWOSM does a data-independent learning rate selection. It is natural that for hard tasks (ImageNet) the data-independent choice may not work well. We note that while DPAdamWOSM does not require tuning the learning rate, we still need to tune the number of epochs. Therefore, even if further tuning $T$ for DPAdamWOSM could match the utility of the linear scaling rule, it would not match the privacy guarantee. Ultimately we think these works are compatible, because we can use our hyperparameter tuning procedure to tune the number of epochs in DPAdamWOSM.
>
> We also provide a quantitative comparison to the RDP tuning method in [4] and find that under their experimental setting, our method is 3.5% better on CIFAR10 for the same privacy cost $\varepsilon=1$. In this experimental setting they do linear probing on a ResNet20 checkpoint pretrained on CIFAR100 and achieve 67% on CIFAR10 at $\varepsilon=1$ (Figure 3, Figure 4 in [4]). We use the linear scaling rule and obtain 70.5% accuracy with $\eta=1, T=50, \varepsilon=0.9$. We have added the code to reproduce this experiment to the codebase linked in the first line of the Appendix (also provided to the AC in another comment as per the rebuttal guidelines). The reason our method outperforms [4] is that we do select $\eta, T$ adaptively and scale them with $\varepsilon$ whereas they only do a random search. [4] is specific to RDP. Our method uses GDP because PLD accounting is known to improve over RDP, but we could just as easily use RDP.
>
> > Unclear why the initial hyperparameter search can be done with such a small privacy budget...
>
> The initial hyperparameter search is itself a random search. However, we are just looking for one combination of hyperparameters that produces nontrivial performance. The intuition is that even if the results for 3 hyperparameter combinations are [2%, 5%, 10%] which are all very bad, the relative ordering between this still hints that the hyperparameter combination that produced 10% still lies on the optimal set of hyperparameters. If we randomly initialize r_0, we may have good performance, or (more likely) we may end up with a bad set of hyperparameters. However, the additional cost of trying a few more candidates for r_0 is negligible since the privacy cost of these runs is so low, approximately $\varepsilon=0.01$.
>
> > How does this method compare with optimization algorithms that reduce the need for hyperparameter tuning?
>
> Could you provide an example so that we can make a comparison?
>
> Bib (from reviewer)
> [1] Kulynych, Bogdan, et al. "What you see is what you get: Distributional generalization for algorithm design in deep learning." arXiv preprint arXiv:2204.03230 (2022): 13. [2] Hulkund, Neha, et al. "Limits of Algorithmic Stability for Distributional Generalization." (2022). [3] Mohapatra, Shubhankar, et al. "The role of adaptive optimizers for honest private hyperparameter selection." Proceedings of the aaai conference on artificial intelligence. Vol. 36. No. 7. 2022 [4] Koskela, Antti, and Tejas Kulkarni. "Practical differentially private hyperparameter tuning with subsampling." arXiv preprint arXiv:2301.11989 (2023).

---

> > ### Author Response · Authors · 2023-08-16
> > **Followup to Weakness #3 / Question #1 and more details on requested comparisons to other hyperparameter tuning methods**
> >
> > Here we provide further details for the 3rd weakness and 1st question as asked by the reviewer. We also provide some more insight on the comparisons we did with [3, 4], omitted from the previous comment due to space constraints.
> >
> > > Unclear why the initial hyperparameter search can be done with such a small privacy budget even though this is a key factor driving the performance of the algorithm.
> >
> > > What is the intuition for why the initial hyperparameter search can be done with such a small privacy budget? Is it possible to simply randomly initialize $r_0$ and achieve similar performance?
> >
> > Please see our comment on the AC’s thread for a comparison to random search.
> >
> > In `linear_scaling.py` we provide code to run the full hyperparameter search routine. In the official comment, we have provided example traces of the searched hyperparameters for CIFAR10. As per the reviewer's recommendation, we include here an analysis of what happens when we increase the privacy budget allotted to the first hyperparameter search from $\varepsilon=0.01$ to $\varepsilon=0.05$, again on CIFAR10. Naturally, allocating more budget to the first hyperparameter search means that we cannot allocate as much privacy budget to the final run. All runs are averaged across 5 independent instantiations of the full hyperparameter search procedure and we report the standard deviation.
> >
> > | $\varepsilon_1$ | $\varepsilon_2$ | $\varepsilon_f$ | Accuracy (%) | Standard Deviation |
> > |------------------|------------------|------------------|--------------|--------------------|
> > | 0.01             | 0.05             | 0.97             | **98.88**        | 0.01               |
> > | 0.05             | 0.1              | 0.96             | 98.85        | 0.03               |
> > | 0.05             | 0.2              | 0.9              | 98.81         | 0.01               |
> >
> > We find that increasing the privacy budget allocated to the first hyperparameter search has only a negligible impact on the final utility for CIFAR10. However, a larger value of $\varepsilon_1$ may serve to stabilize the initial phase of hyperparameter search for more difficult tasks. As guidance for practitioners, we still suggest that the hyperparameter search be instantiated with a small value of $\varepsilon_1$, and if the results are no better than random chance, we can terminate the search and increase $\varepsilon_1$ by, e.g., a factor of 5 or 10 until the results become nontrivial. In the worst case, we waste only $\varepsilon_1=0.01$ or some other small privacy budget due to fully adaptive composition.
> >
> > **More details on comparisons with [3,4]**
> >
> > Our goal in these comparisons is to do a fair or “apples-to-apples” comparison. For [3], that means that we reimplemented their method (because they did not provide code and did not evaluate on any of the same benchmarks that we do) in `wosm_impl.py` and gave it all the optimal hyperparameter choices that we provide our methods, such as the full-batch GD setting. We are confident that the comparison to [3] is fair, and even then we do not make [3] pay for the cost of tuning the number of epochs hyperparameter in that evaluation.
> >
> > For 4] (the paper that is more concurrent to ours, with their latest revision in June 2023), we do a fair comparison by running our procedure in the exact setting they report results on. Our methods are similar in that we both run hyperparameter tuning on a set of candidates. We tune the learning rate and number of epochs, leaving the batch size as the full batch setting. They tune the learning rate, number of iterations, and the subsampling ratio $\gamma$ and find that the best values of $\gamma$ are around $0.02$. In contrast, we do our experiments in the full batch setting (batch size of 50000 for CIFAR10).
> >
> > To analyze why our method outperforms [4], we look to the adaptive nature of our hyperparameter search and the very small privacy cost we pay. As noted above, the privacy cost we are paying is the difference in performance of the final run between $\varepsilon=0.97$ and $\varepsilon=1$. Therefore our hyperparameter tuning is very privacy efficient, because we are able to take advantage of the superior GDP composition. By contrast [4] relies on RDP which is known to be suboptimal and they allocate more privacy budget to tuning according. This is because they just do random search. However, our search is adaptive in that we have a prior that the learning rate and iterations increase with $\varepsilon$.
> >
> > We can further compare how close their final searched hyperparameters are to what our method finds, because we use the same model optimizer etc. We can look at their results from Fig. 5 where they tune learning rate; they find the same epochs as us. Their original learning rate of $\eta=0.15$ is too small, but they use subsampling. We can apply the original linear scaling rule to scale up their optimal learning rate to $\eta = 0.15 \times \frac{1}{\gamma} = 0.15 \times \frac{1}{0.02} = 7.5$. This is too large.

---

> > > ### Comment · Reviewer_BCCV · 2023-08-21
> > > **Apologies for late response.**
> > >
> > > I apologize to the authors for the late response to their rebuttal. I appreciate the detail taken in crafting their response. I thank the authors for providing these comparisons to the existing baselines and taking feedback from reviewer kJWb to ensure they are an apples to apples comparison. Based on the overall rebuttal, these results, and rebuttals to other reviewers I will keep my score as is. There are interesting contributions in this paper on training DP models and hyperparameter tuning but I do feel that given many of the pretrained models seem to be powering the performance and it seems that some of the tasks are simply robust to the selection of learning rate the wide applicability is less clear.

---

> > > > ### Author Response · Authors · 2023-08-21
> > > >
> > > > We thank the reviewer for their response. Indeed, our paper focuses on the problem of maximizing performance from pretrained models. However, even when the models are pretrained there can still be room for improvement, as we showed in the comparison to [4] where our model improves in performance over a very recent work by $3.5$%. We feel that our empirical evaluation across 20 tasks in the main paper, including both CV and NLP, linear probing and full fine-tuning of transformers, across multiple types of domain shifts, is sufficient to validate the wide applicability of our paper. If there are any tasks you feel we left out of our evaluation, please let us know and we can add the experiments for the camera ready.
> > > >
> > > > Once again we appreciate the reviewer's care in selecting papers to request additional experiments for. We are confident that the camera ready for our paper is strengthened by adding in these comparisons.

---

### Official Review · Reviewer_tXcJ · 2023-07-27

**Soundness:** 2 fair
**Presentation:** 1 poor
**Contribution:** 3 good
**Rating:** 5
**Confidence:** 3

**Summary:**

This paper proposes a new method to conduct hyper parameter tuning for DP stochastic gradient descent. The method is based on a linear scaling rule, with two pilot runs using small PLBs and a third run chosen based on a linear extrapolation from the first two. The pilot runs are used to establish an estimate of the interpret and the slope that the total step size r would have with the PLB. The author uses this linear scaling rules to demonstrate that it works as well as grid search in optimizing for the accuracy in a suite of benchmark tasks, and attempts to apply this rule to perform empirical analysis on the potential of making existing model architectures DP and the issue of robustness against domain shifts.

My assessment, consisting of strengths, weaknesses, and questions, can be found in the sections below.



**Strengths:**

The best thing about this paper is that it develops a method based on an intuition that is potentially worthwhile. This intuition is captured in the small paragraph in Section 2, titled Linear Scaling is Intuitive. What the authors have proposed is essentially a dimensional reduction to the hyperparameter search, and the reason why that works, in the sense that what you end up finding may not be so far off from a greedier search, is due to the geometry where you force the updates to be more congruent with each other. The whole idea of a linear scaling would otherwise be rather unremarkable, but if the author can further develop this intuition, formalize it and expand on it, it would contribute some insight to the literature.

**Weaknesses:**

The most damning weakness of this paper is that it is written without due care. As a consequence, the main results and the accompanying algorithm are not correct as stated. I don’t suggest that the author is not capable of presenting the correct science -- to that question I do not know the answer. However, as things stand, the paper is not ready to be published.

The presentation in the introductory and main result sections wanders seemingly fluidly between epsilon-DP, (epsilon, delta)-DP and Gaussian DP:
1. Definition 1.1 is given in the language of (epsilon, delta)-DP;
2. The DP-SGD Definition is given without a quantification of its DP guarantee at all;
3.  Algorithm 1, which employs the DP-SGD given before, states that its output is epsilon-DP, where an alleged PLB accounting between epsilon and sigma is not supplied. (In reality, a delta would be needed, so the provided guarantee is incorrect to begin with.)
4. Then Proposition 2.1, which concerns Algorithm 1, gives a GDP guarantee in relation to sigma only, where sigma is not constructed as a function of epsilon (or the missing delta) in Algorithm 1;
5. Corollary 2.2 now qualifies Algorithm 1 as (epsilon, delta)-DP, with a one line proof given in the Appendix citing another work and has no substance on its own.

All of the above is confusing at best. For a standard reader, a student coming into the DP world for example, these are not pedagogically informative.

Back to Algorithm 1:
1. It contains four privacy loss budget expressions: epsilon, epsilon_0, epsilon_1, and epsilon_f. Based on the context, am I to infer that epsilon is the sum of the rest of the three?
2. The quantity r on the 12th line (beginning with Decompose). Is this a generic r, as you use it on line 7, or is it in fact referring to r* on line 9?
3. When you speak of the “decomposition” or r, what is to be found exactly -- eta given r and T (my guess), T given r and eta (please explain), or both eta and T given r (please explain as well)? If my guess is correct, then do we know that the eta found here will automatically satisfy the condition given in Theorem 2.3?

Line 143 begins with “We apply this theorem to logistic regression.” Then Line 151 continues, “While our theorem only holds for linear models…”. Nothing said between Line 143 and Line 151 constitutes a proof that Theorem 2.3 applies to linear models. This point should either be rectified with a formal analysis or deleted, so as to not be an exaggeration of contribution.

Section 3.1 is misleading and should be thoroughly rewritten to rid all expressions of “randomly”, “sample”, and “uniformly”. The author picked the experimental values. No sampling, particularly random sampling nor uniform random sampling of values took place. It is not clear to what is “r = 75” an approximation (Line 179).

In addition, based on my reading of Section 3.2 I believe it should not be presented as is.  My understanding of what Section 3.2 does is that it uses the linear scaling rule proposed in this work to construct "accuracy hypotheticals” for the listed models and datasets as well as the domain shift situations, and compare those numbers with existing experimental results. If that is the case, this is a dangerous operation. The linear scaling rule, when used as a heuristic to make tuning faster, is fine as the worst that could happen is that one misses out on the most efficient model tuning. However, the way that the rule is employed in Section 3.2 it is taken as a scientific theory between epsilon and accuracy. The accuracy numbers you get from it is no different than a terribly extrapolated number based on a linear model fitted with two data points. If you really want to use the linear scaling rule to poke at the said questions, actual experiments should be conducted to confirm these extrapolations. Of course, I may have misunderstood what was actually done and in particular, whether actual experiments were performed — although if so, what would be the contribution from the linear scaling rule?


**Questions:**

Below are a set of minor comments.

1. Please define SOTA.
2. Please also define r, eta, and T before they are used for the first time. After the first time, there is no need to always state that one is the product of the other two.
3. Lines 83-85: I understand that these are conclusions based on your empirical analysis in Sections 3.2 and 3.3. As the statements read, they cannot possibly be correct without qualifications to the specific characteristics of the models and benchmarks that you examine.
4. The fourth to last line in Algorithm 4 does not read properly.
5. Line 106: “We provide a privacy guarantee in 2.” What is 2?
6. The x axes of Figure 3 are not labeled.
7. Line 196: “…their value of r is ≈ 1000× smaller than ours.” What you mean is that “their value of r is about a thousand times smaller than ours.”
8. Line 216: “In Fig. 3 we report that following Algorithm 1 produces new state-of-the-art results for all values of ε, shown in Table 5.” This sentence seems to imply that you have applied Algorithm 1 to all Models and datasets listed in Figure 5 (which should be Table 5). This is contrary to my understanding that Section 3.2 is a thought experiment.


**Limitations:**

As stated before, I believe the paper is written hastily to the point that the central results presented are incorrect, significantly harming the quality of the contribution and its readability. I am also concerned with the scientific merit of Section 3.2. These points are elaborated in detail in my comment section on Weaknesses.

---

> ### Author Rebuttal · Authors · 2023-08-01
>
> We thank the reviewer for their detailed review; in this rebuttal we will clear up major misunderstandings and provide clarifications.
>
> The reviewer has commented that they believe our experimental analysis in Section 3.2 is a 'thought experiment' and that we did not run all the experiments. We wish to clarify the reviewer's misunderstanding -we actually ran all of the experiments. The contribution is that running these experiments with the linear scaling rule allows us to pick hyperparameters with very high utility that also accounts for the privacy cost. In comparison, prior approaches produce suboptimal hyperparameters (see e.g., the comparison between Bu et al and ours) and also don't account for the privacy cost of hyperparameter tuning. We provide extensive experimental validation for this in Section 3 and Appendix A.2 and code to reproduce all our results in the first line of the Appendix. Figure 3 corresponds to one row of the table in Figure 5. The contribution as stated in the caption of Figure 5 is that the privacy cost of running the experiment would include the privacy cost of tuning the hyperparameters because of the linear scaling rule. Ordinarily doing a grid search would have a very large privacy cost. The optimality gap between the linear scaling rule and grid search is given by Figure 3. As you noted, "the linear scaling rule [can be used] as a heuristic to make tuning faster". It is the goal of our work to make tuning faster and more private, and our experimental analysis in Section 3 shows that we succeed on both accounts.
>
> We appreciate the reviewer's careful reading of our theory in Section 2. We have incorporated these comments into the following overhaul. First we will rewrite Lines 125-134 and update the corresponding lines in Alg. 1 with the following procedure. Given a desired final $(\epsilon, \delta)$-guarantee, we will use the GDP-approx DP conversion from Corollary 2.13 in [18] to find the appropriate value of the parameter $\mu$ for GDP. Then we will allocate $\mu$ across the hyperparameter optimization runs and final run according to $\mu = \sqrt{3 \mu_{1}^{2} + 3 \mu_{2}^{2} + \mu_{f}^{2}}$, where $\mu_{f}$ is the privacy parameter for the final run that uses the hyperparameters adaptively chosen by using the linear scaling rule on the outputs of the runs with smaller privacy budgets $\mu_{1}, \mu_{2}$. Then for a given $\mu_{i}$, we will decompose this into $(T, \sigma)$ according to Proposition 2.1 in the main paper.
> By writing the privacy analysis in this manner, we can stay in GDP the entire time and we hope these presentation changes will address the reviewer's feedback.
>
> We now address line by line weaknesses and comments. In particular, we show that our Theorem does have a proof that applies to linear models.
>
> > The quantity r on the 12th line (beginning with Decompose). Is this a generic r, as you use it on line 7, or is it in fact referring to r* on line 9?
>
> You are correct; this is referring to r* on line 9. We will fix this.
>
> > When you speak...
>
> Both eta and T given r; the function we use is just a simple nested for loop that iterates over randomly shuffled arrays of valid epoch and learning rate values and checks whether their product is within some tolerance of the given r. As you noted, it's likely that the learning rate does not satisfy the condition in Theorem 2.3, because the smoothness constant has not been calculated. We will clarify this in the camera ready. We will provide a subroutine for decomposing r in the camera ready.
>
> > Line 143...
>
> The proof of Theorem 2.3 can be found in Appendix A.5, as referenced on line 129. The exact line for the proof of Theorem 2.3 starts on line 855.
>
> > Section 3.1...
>
> We agree that without attestation for the random number generation we should omit any mention of it from the text. However, we did not cherry pick random numbers, there are a wide range of hyperparameters which produce extremely high performance on CIFAR10. If the reviewer doubts our claims, we invite the reviewer to use our provided code to evaluate some random combinations of hyperparameters to gauge the robustness, or look at our Figures 19/20. Line 179 omits the bias term (+1.25) because 76.25 would require using a learning rate of 0.7625 which is nonstandard. By nonstandard, we mean that our code has two arrays, valid_epochs and valid_lrs, and randomly shuffles these. It then iterates until a combination that is within a tolerance of r is found; in this case, 0.75 * 100 is within the tolerance (as would be 1 * 75).
>
> > Please define SOTA.
>
> We write out state-of-the-art in line 9 and will add a "(SOTA)" afterwards so that it serves as a definition
>
> > Lines 83-85...
>
> We will add the qualifiers "when the models are pretrained from public data" and "on the spread of 20 benchmarks we examine"
>
> > The fourth to last line in Algorithm 4 does not read properly.
>
> In this line we are substituting the unit norm clipping threshold for the DP-SGD equation and also saying we always do the full-batch update. We will explicitly say this in the text of Algorithm 1.
>
> > Line 106...
>
> Apologies, accidentally used \ref instead of \cref. This should refer to Proposition 2.1
>
> > The x axes of Figure 3 are not labeled.
>
> Apologies, the x-axis is epsilon, we will fix this.
>
> > Line 196...
>
> Yes, that's correct, thank you.
>
> > Line 216...
>
> Section 3.2 is not a thought experiment. We carried out all the experiments. You are correct that it should be Table 5; we will fix this.
>
> > As stated before...
>
> We hope that our clarifications (primarily on the misunderstanding that Section 3.2 is a thought experiment) can improve your score. We thank you for the detailed review.

---

> > ### Comment · Reviewer_tXcJ · 2023-08-11
> >
> > I thank the authors for taking their time to respond. I have updated my rating in a favorable manner to acknowledge the attempt they have made to clarify the conceptual understanding.
> >
> > I will say one more thing for the authors's own benefit. Moving forward, please do not make statements such as "If the reviewer doubts our claims, we invite the reviewer to use our provided code to evaluate..." You are not in an antagonistic relationship with the reviewers, who are merely speaking on behalf of potential readers -- consumers of the science which you are trying to produce. The onus is on you to demonstrate trustworthy work, not on the rest of the world to prove it otherwise.
> >
> > This paper still has a long way to go before it can be called a good piece of literature. I have no doubt that the authors have done the hard work producing the experimental results. However, presenting the work in a way that a reader can, without your clarifying on their side, understand the correct messages is part of the job, arguably just as hard.

---

> > > ### Author Response · Authors · 2023-08-14
> > >
> > > We would like to apologize that our previous comment seemed antagonistic, as that was not our intention. We merely wished to bring to the reviewers' attention that we have included code, which may serve as supplementary evidence. We appreciate the reviewers' diligent efforts in providing insightful comments that are helpful in enhancing the quality of our paper. In the forthcoming revision, we will craft a more persuasive discourse to assure that our results do not seem cherrypicked or too good to be true. We will incorporate all of the reviewer's feedback to ensure our work is trustworthy and displays the correct message. We also welcome any further feedback or reservations the reviewer may still hold.

---

### Comment · Area_Chair_M9PS · 2023-08-15
**Updated code repository and followups**

Dear all,

The authors have shared an updated link to an anonymous code repository, given that the one from the original supplementary file has expired: https://anonymous.4open.science/r/dp-custom-32B9/. Authors: please feel free to add any additional information to this thread on how to run the code, etc.

To summarize why the updated code is relevant, reviewers KKeT and kJWb both find the reported accuracy high (e.g., KKeT writes "the paper under submission seemed to present implausibly strong results to me" and kJWb writes "I somehow find it hard to believe that you would get 99 percent test accuracy for CIFAR-10 with epsilon = 1").

I tend to agree with the reviewers that the paper reports surprisingly good results. I also especially agree with reviewer KKeT's argument that, all else equal, non-private parameter tuning should beat or match any private parameter tuning approach. For both CIFAR-10 and CIFAR-100, the epsilon = 1 results for the proposed method beats the epsilon = infinity results for all prior work. Regarding this, the authors write "We obtain better results even compared to the epsilon = infinity numbers in other papers via a combination of better models and better hyperparameters."

Authors: can you also point me to the place within your code repository that actually does the hyperparameter tuning? As far as I can tell, hyperparameters are specified via command line arguments.

Thanks,
Your AC.

---

> ### Comment · Reviewer_kJWb · 2023-08-15
>
> Thanks. I had a quick look at the code and I'd also like to ask: is it possible/easy to get similar results by using e.g. Opacus? A library called "FastDP" seems to be used for the gradient evaluations.

---

> > ### Author Response · Authors · 2023-08-15
> >
> > We thank the reviewer for the response. Opacus can be used instead of fastDP to get the same results. We have just pushed an update to the code such that passing '--privacy_engine opacus' will run with Opacus instead of fastDP, so that installing fastDP is not necessary (see our response to the parent comment for the source of fastDP).

---

> ### Author Response · Authors · 2023-08-15
> **Instructions for running the code**
>
> **We agree that "all else equal, non-private parameter tuning should beat or match any private parameter tuning approach" but critically note that not all other parameters are equal. In particular, we use better pretrained models than prior work. We now detail how to run our code and reiterate the explanations for why we beat prior work. In the forthcoming revisions, we will craft a more persuasive discourse to assure that our results are not presented as 'too good to be true'.**
>
> **Better pretrained models:**
> Prior works that obtained SOTA results use a range of different pretrained models. For example, [7] uses beit, [51] uses different sizes of ViT, and [15] uses WRN-40-4. In Figure 5 we provide results with different pretrained models. The best of these, beitv2, is reported in Table 2. When we provide numbers from other papers they are always quoted from those papers unless the compared value of $\varepsilon$ is not reported in that paper. In the repo we have already provided extracted features from beitv2 (hence the somewhat large file size) for convenience.
>
> **Better hyperparameters:**
> Our method tunes the primary hyperparameters of optimization, learning rate and number of iterations, but a number of other hyperparameters exist that impact accuracy. We briefly summarize how to set these hyperparameters in Line 94. In Table 1, Appendix A.2, we quantify the impact of setting each hyperparameter and find that this improves accuracy significantly. The rest of Appendix A.2 describes each of these hyperparameter choices in detail and provides ablations. We emphasize that it **is** possible to have worse performance with larger $\varepsilon$ if these hyperparameters are worse. *Note that even at $\varepsilon=\infty$ we still employ per-sample gradient clipping, as prior work does.* This can be verified by running our code without using momentum (as is the case in the highly-cited [15]):
> ```
> python finetune_classifier_dp.py --epsilon 0.0 --momentum 0.0
> Mean accuracy 98.72
> ```
>
> **Running our Code:**
> Running `python finetune_classifier_dp.py` will run fine-tuning on the extracted features from CIFAR10 (that we have uploaded for convenience) with $\varepsilon=1.0$; this should run in just a few minutes on a single GPU and can even be run in a Colab notebook. The dependencies are listed in the README. The only nonstandard dependency is fastDP, an AWS repo that can be installed from https://github.com/awslabs/fast-differential-privacy/tree/main.
>
> **Reproducing our Results:**
> This is our output from running 10 runs with the command `python finetune_classifier_dp.py --num_runs 10`; where the randomness is over the random seed that generates the added noise. We will revise the numbers in Figure 2 and other figures in the main body to include the mean accuracy and error bars for the camera ready; we were unable to do so for the main paper because of the sheer number of experiments we report.
> ```
> Mean accuracy 98.90
> ALL ACCS [99.01, 98.9, 98.88, 98.86, 98.89, 98.83, 98.95, 98.9, 98.94, 98.88]
> Std accuracy 0.05
> ```
> Some default hyperparameters are provided for convenience and other hyperparameters can be passed on the command line (for a full list of hyperparameters check utils.py) for example we can run $\varepsilon=\infty$ with the command `python finetune_classifier_dp.py --epsilon 0 --epochs 200`; where there is no randomness because the model weights are initialized at zero and we do full-batch gradient descent
> ```
> Mean accuracy 98.99
> ```

---

> ### Author Response · Authors · 2023-08-15
> **Instructions for running linear scaling**
>
> We hope that by providing instructions for running the code, and in particular, explaining one of the major differences between prior work and our work in the different pretrained models, we clarify our approach. The main contribution of our paper is not in using these better models, and so we have clearly underexplained this point. In forthcoming revisions, we will make sure to discuss the impact of pretrained models in more detail. However, the main contribution of our paper is still our proposed hyperparameter tuning approach, which we feel is novel and provides good results. In this comment, we provide instructions for running our proposed approach, the linear scaling rule.
>
> Running the linear scaling rule for hyperparameter tuning with the default beitv2 embeddings and the CIFAR10 dataset can be done with `python linear_scaling.py`. Below, we provide our output from running `python linear_scaling.py` three times. The searched hyperparameters and optimal hyperparameters will vary, as will the final accuracy. For the camera-ready version, we will include error bars for the main results.
>
> First run:
> ```
> Final accuracy: 98.89
> All searched etas: [0.9, 0.1, 0.1, 0.9, 0.95, 0.9, 0.95]
> All searched ts: [20, 105, 150, 45, 30, 45, 130]
> All searched sigmas: [1145.032, 2620.688, 3149.775, 207.984, 169.683, 207.984, 43.758]
> All searched accs: [76.76, 71.09, 68.6, 98.27, 98.29, 97.99, 98.89]
> Total privacy cost for final accuracy including the privacy cost of hyperparameter search  0.993
> ```
> Second run:
> ```
> Final accuracy: 98.89
> All searched etas: [0.95, 0.15, 0.25, 0.50, 0.3, 0.35, 0.7]
> All searched ts: [10, 85, 40, 50, 50, 50, 85]
> All searched sigmas: [809.665, 2360.260, 1619.316, 219.354, 219.354, 219.354, 35.384]
> All searched accs: [77.69, 70.01, 78.37, 98.28, 98.36, 98.35, 98.89]
> Total privacy cost for final accuracy including the privacy cost of hyperparameter search  0.993
> ```
> Third run:
> ```
> Final accuracy: 98.89
> All searched etas: 0.20, 0.05, 0.80, 0.60, 0.35, 0.30, 1.00
> All searched ts: [30, 80, 5, 40, 85, 50, 150]
> All searched sigmas: 1401.282, 2290.064, 572.511, 196.118, 285.879, 219.355, 47.006
> All searched accs: [81.86, 80.67, 80.45, 98.33, 98.16, 98.3, 98.89]
> Total privacy cost for final accuracy including the privacy cost of hyperparameter search  0.993
> ```
>
> The relevant constants at the top of `linear_scaling.py` dictate the privacy budget allocated to the hyperparameter search vs the final run. Note the use of GDP composition that is the key to doing the hyperparameter search with a very small privacy cost. For example, doing 3 runs with $\varepsilon_1=0.05$, 3 runs with $\varepsilon_2=0.1$, and one run with $\varepsilon=0.96$ will give a composed privacy cost of $\varepsilon_f=1$. That is, the cost of finding better hyperparameters is only the marginal utility gained by increasing the privacy budget from $\varepsilon=0.96 \rightarrow \varepsilon=1.0$. We provide code dedicated to the privacy analysis in `test_prv.py`. Naturally, altering these constants has an impact on the utility of our method. However, as long as $\varepsilon_f$ is large enough ($>0.9$), these variations are small on the order of $0.1 \%$ accuracy. We will provide a more detailed analysis of the privacy budget allocation in the camera ready. Here is an ablation.
>
> | $\varepsilon_1$ | $\varepsilon_2$ | $\varepsilon_f$ | Accuracy (%) | Standard Deviation |
> |------------------|------------------|------------------|--------------|--------------------|
> | 0.01             | 0.05             | 0.97             | **98.88**        | 0.01               |
> | 0.05             | 0.1              | 0.96             | 98.85        | 0.03               |
> | 0.05             | 0.2              | 0.9              | 98.81         | 0.01               |
>
> Please let us know if there are any issues with running the code. Our apologies for the delay; we hope that there is sufficient time in the discussion period for the reviewers to experiment with the provided code. We understand that the reproducibility is a major factor, and will make sure that more text is given over to discussing this in the camera ready.

---

> > ### Comment · Reviewer_kJWb · 2023-08-16
> >
> > Thanks for the replies! I did not run the code but I had a closer look at it. So this is my impression:
> >
> > The paper uses the AWS library "fast-differential-privacy":
> > https://github.com/awslabs/fast-differential-privacy/tree/main
> >
> > This library is based on a line of work by Bu et al. (e.g., a NeurIPS2022 paper). They also report these high accuracies for fine-tuning of vision transformers, see e.g. Tables 7 and 15 and Figure 5 here:  https://arxiv.org/pdf/2210.00036.pdf.   They also report 99% test accuracy for CIFAR-10 for eps=1.0 (Table 15). Looking at Figure 5 and rest of the results, I get the impression that the pretrained models are so powerful that tiny fine-tuning suffices.
> >
> > I wasn't aware of this line of work (will definitely have a closer look), but in any case it looks like the reported results here are in line with this existing work. And that there is perhaps small contribution coming from the additional tricks mentioned in the appendix.
> >
> > I also looked at the hyperparameter search part of the code and it looks ok to me. All the candidate models are released in the process and GDP composition results are used (full batch training). I found it a little bit suprising that 3 randomly chosen candidates (for both eps=0.01 and 0.1) are sufficient to consistently obtain good learning rates using the scaling. But perhaps the tasks are then not so sensitive to the choice of the learning rate.
> >
> > Then comes the question, what is really the contribution of this paper? Looking at your Figure 5 (The table), it seems that for CIFAR-100 you have the biggest drop in accuracy compared to non-DP. Also, looking at the heat maps in the appendix (Figure 19 and 20), it looks like CIFAR-100 models are clearly the most sensitive to the choice of the learning rate and that the other models are not that sensitive. This would make me think that the proposed algorithm is perhaps not really finding the optimal learning rate values in the difficult cases, and that a wild guess / default value could give as good results.
> >
> > I believe there is contribution in the proposed training tricks which are given in the appendix. However, there also, I think it would make sense to only report comparisons against the results of Bu et al. (see e.g. Ablation of Appendix A.2) since they already report such big improvements to the previous work. Anyways, currently the main story of the paper seems to be this linear scaling rule of the learning rate for fine-tuning of large models and I am quite unsure about the value the proposed method.

---

> > > ### Author Response · Authors · 2023-08-16
> > > **Contributions of the linear scaling rule over other hyperparameter search baselines**
> > >
> > > Thank you for the detailed reply. In the main paper we have included a comparison between the linear scaling rule and the oracle (non-private grid search) in Figures 3/8. In the response to reviewer BCCV we have also included comparisons to two other DP methods that aim to find the learning rate. In this response we are including a comparison to random search. The camera ready will contain all these comparisons.
> > >
> > > We are glad that you find our results are in line with prior work. As you have noticed, some of our comparisons (Table 1 in Appendix A.2, Figure 3 in the main paper) are done on the beit model for CIFAR100. This is the same model as the prior work you cited and a more challenging dataset than CIFAR10. We will take into account your suggestion and only provide the ablations with reference to the existing baseline; however, note that we only do linear probing for those CV tasks whereas the paper you cited does full fine-tuning and the BiTFiT method (https://arxiv.org/abs/2106.10199).
> > >
> > > Secondly we can turn to the natural question, that is "Could a wild guess give as good results as the linear scaling rule", which we are interpreting as random search. If you had something else in mind, please let us know and we will provide that comparison.
> > >
> > > We can measure the improvement of the linear scaling rule over random search by running linear scaling a number of times and comparing this to random search (that does not pay any privacy cost for hyperparameter tuning and instead just picks a random value from the same grid). Our method gets $89.32$% and outperforms random search which gets $88.19$%. This is very small in absolute terms, but the relative error rate reduction is $69.75$%. We calculate this as $\dfrac{(best - linear) - (best - rand)}{best-rand}$, where the best result at $\varepsilon=1$ is $89.81$%.
> > >
> > > We did not report this comparison in the paper because in Figure 3 we compare our linear scaling rule on CIFAR100 with beit to the 'upper bound', that is grid search where we _do not consider the privacy cost of hyperparameter search._ That is, the line marked 'grid search' in Fig. 3 only reports the best result across the grid for a given $\varepsilon$; ex: $89.81$ at $\varepsilon=1$. Naturally, it is impossible for random search to outperform this 'grid search upper bound'. We will add a line for random search on Figure 3; as we will now show, it will be worse than both grid search and linear scaling.
> > >
> > > The performance of the linear scaling rule averaged over 5 runs is $89.32$ with a standard deviation of $0.56$; here are the outputs for each of the 5 runs.
> > >
> > > ```
> > > Final accuracy: 89.7
> > > All searched etas: 0.10, 0.30, 0.05, 0.60, 0.40, 0.15, 0.90
> > > All searched ts: [115, 20, 145, 40, 45, 60, 130]
> > > All searched sigmas: 2743.32, 1145.03, 3087.53, 196.12, 207.98, 239.97, 43.76
> > > All searched accs: [2.91, 2.84, 3.21, 60.77, 60.95, 59.47, 89.7]
> > > Total privacy cost for final accuracy including the privacy cost of hyperparameter search  0.994
> > >
> > > Final accuracy: 89.17
> > > All searched etas: 0.10, 0.40, 0.25, 0.30, 0.90, 0.20, 0.70
> > > All searched ts: [115, 30, 55, 55, 15, 95, 60]
> > > All searched sigmas: 2743.32, 1401.28, 1896.96, 229.70, 119.98, 302.38, 29.73
> > > All searched accs: [1.97, 1.5, 3.71, 63.12, 56.93, 61.6, 89.17]
> > > Total privacy cost for final accuracy including the privacy cost of hyperparameter search  0.994
> > >
> > > Final accuracy: 89.62
> > > All searched etas: 0.55, 0.20, 0.05, 0.30, 0.55, 0.20, 0.95
> > > All searched ts: [15, 10, 60, 75, 35, 80, 145]
> > > All searched sigmas: 990.85, 809.67, 1981.71, 268.80, 183.52, 277.35, 46.22
> > > All searched accs: [2.62, 2.79, 3.08, 61.31, 60.51, 63.99, 89.62]
> > > Total privacy cost for final accuracy including the privacy cost of hyperparameter search 0.994
> > >
> > > Final accuracy: 89.82
> > > All searched etas: 0.20, 0.05, 0.10, 0.45, 0.55, 0.20, 0.95
> > > All searched ts: [50, 140, 20, 40, 40, 95, 105]
> > > All searched sigmas: 1811.23, 3034.58, 1145.03, 196.12, 196.12, 302.38, 39.33
> > > All searched accs: [2.9, 2.17, 2.14, 60.14, 60.95, 63.57, 89.82]
> > > Total privacy cost for final accuracy including the privacy cost of hyperparameter search 0.994
> > >
> > > Final accuracy: 88.3
> > > All searched etas: 0.95, 0.30, 0.70, 0.20, 0.20, 0.15, 0.40
> > > All searched ts: [5, 35, 10, 50, 105, 80, 65]
> > > All searched sigmas: 572.51, 1517.29, 809.67, 219.35, 317.88, 277.35, 30.94
> > > All searched accs: [3.08, 3.25, 2.76, 58.03, 62.1, 62.9, 88.3]
> > > Total privacy cost for final accuracy including the privacy cost of hyperparameter search 0.994
> > > ```
> > >
> > > This is the output of random search performed 10 times;
> > > ```
> > > All searched etas: 0.40, 0.55, 0.70, 0.10, 0.15, 0.35, 0.05, 0.30, 0.05, 0.65
> > > All searched ts: [40, 130, 110, 55, 20, 80, 110, 75, 100, 95]
> > > All searched sigmas: 23.61, 42.56, 39.15, 27.68, 16.69, 33.39, 39.15, 32.33, 37.33, 36.38
> > > All searched accs: [87.55, 89.78, 89.67, 87.14, 86.08, 88.71, 87.41, 88.35, 87.53, 89.66]
> > > Mean and standard deviation of accs: 88.19 +- 1.19
> > > ```
> > > In addition to being worse on average than linear scaling, the standard deviation is quite high.

---

> > > ### Author Response · Authors · 2023-08-16
> > > **(cont.) Response to Reviewer kJWb**
> > >
> > > (cont. from previous comment due to char. limit)
> > >
> > > We first clarify the timing of citations, because there are various (Bu et al) papers. The NeurIPS 2022 paper that introduces the fastDP library is "Scalable and Efficient Training of Large Convolutional Neural Networks with Differential Privacy". FastDP is augmented with an ICML 2023 paper "Differentially Private Optimization on Large Model at Small Cost"; this introduces better book-keeping (not keeping around gradients in memory unnecessarily) as detailed in the fastDP file `supported_layers_grad_samplers.py` by passing the flag `clipping_mode="MixOpt"`. The paper you have cited, "Differentially Private Bias-Term only Fine-tuning of Foundation Models" is an arXiv preprint and enables using the BiTFiT method in fastDP by passing the flag `bias_only=True` to only optimize the bias parameters rather than all the parameters. We are not discounting the results of this line of work in any way; we have used the open-source fastDP repository to great effect, and they use the same pretrained models from timm that we do. Hereafter we will refer to these as [BuNeurIPS2022, BuICML2023, BuArxiv] to avoid confusion.
> > >
> > > In our paper we provided a comparison to [BuNeurIPS2022], and as the reviewer has noted, better numbers are available in [BuArxiv]. We will add a comparison to these in the next revision; their numbers are as good as ours. **However, there are two important notes to be made here.**
> > >
> > > First note that while all our CV results use linear probing (that is, we extract features from a model and then train a linear layer on top of those features), [BuArxiv] proposes a parameter-efficient fine-tuning (PEFT) approach that is considerably more complicated and more time/space intensive than linear probing. They update all parameters of the model (for a number of epochs that is a hyperparameter in their method), and then apply BiTFiT (https://arxiv.org/abs/2106.10199). Their best result on beit-large at $\varepsilon=1$ is for the column labeled "2+BiTFiT", meaning they do 2 epochs of full fine-tuning and then run BiTFiT. It is natural that this will outperform linear probing, which is just the baseline PEFT approach. However, there are downsides to this approach. There is no way that we could perform the full fine-tuning portion of their method on, ex, the ViT-g for ImageNet with our compute resources. **The compute efficiency of the linear scaling rule with linear probing (the setting where our Theorem 2 holds) unlocks SOTA results across a wide range of tasks even when compute is too limited to do grid search or full fine-tune ViTs.**
> > >
> > > Second note that, as with other prior works we compare to, [BuArxiv] does not account for the privacy cost of hyperparameter tuning. We believe this is an important problem, and we hope the reviewer agrees. **We emphasize that our proposed method enables generating SOTA results while accounting for the privacy cost of hyperparameter tuning, which is something that no other work has done. A potential lasting contribution of our work is that future work may adopt our linear scaling rule to account for the privacy cost of tuning hyperparameters, instead of using the time-consuming and privacy-intensive grid search or any other hyperparameter search baselines we have compared to in our paper.**
> > >
> > > >Currently the main story
> > >
> > > As described in our Introduction: A common problem that DP researchers face is that there are so many hyperparameters (learning rate, iterations, clipping norm, batch size, optimizer, momentum, weight initialization, updateable weights) that behave so differently from non-private training (increasing iterations increases the noise multiplier, updating more weights decreases the signal to noise ratio, decreasing batch size increases the signal to noise ratio, increasing the clipping norm means adding more noise) that the only option to get "SOTA" results is to do a grid search from scratch, or use the hyperparameters searched by previous work on that same dataset. Indeed, we found that previous work used hundreds of trials of hyperparameter search (Line 188 in our paper). Because of the increased number of hyperparameters and the additional time complexity of DP-SGD over SGD, this grid search is very compute intensive and also has large privacy cost if properly accounted for.
> > >
> > > The main story of our paper is **our linear scaling rule reduces the compute and privacy costs of grid search by an order of magnitude without compromising accuracy across 20 fine-tuning tasks.** The conclusion of this story is: researchers using our method can accelerate the pace of research by reducing the compute needed to produce good results, and address the open question of accounting for the privacy cost of hyperparameter tuning, whether they are doing linear probing on CV (with domain shifts) or full fine-tuning of transformers on NLP. We ultimately believe that if our method were adopted, it would be a big step for DP fine-tuning research.

---

> > > > ### Comment · Reviewer_kJWb · 2023-08-16
> > > >
> > > > Thank you. It seems to me that in this particular example you get the best results for the largest $\eta$'s, and that the random guessing seems to work quite well. The $\eta$-grid, just like in the examples mentioned in the appendix, feels a big hand picked. Likee here, in Tables 5 and 6 of the Appendix, for example, you all the learning rate values are of the same order, for a given value of $\epsilon$. Wouldn't you in practice have a logarithmic grid over several order of magnitude? Then I get the impression you'd need a finer resolution for the grid and your tuning cost would increase.
> > > >
> > > > As I wrote related to the review of BCCV, the comparisons to those mentioned papers [3] and [4] do not seem necessarily fair since they have small batches as hyperparameter candidates whereas you are running full batch DP-GD. I mean you should not only look at the final test accuracies, but you should compare in the same setting. Looking at your results, if you would take the method of [3] or [4] with this learning rate grid for Cifar10 or Cifar100, I believe you could get similar results.
> > > >
> > > > Looking at Appendix B, it seems that you are not using full batches for the language model fine-tuning. How do you then carry out the GDP accounting?

---

> > > > > ### Author Response · Authors · 2023-08-16
> > > > > **Response to Reviewer kJWb**
> > > > >
> > > > > Thank you for your comment; we hope our responses so far have clarified your concerns. In this response we respond to your concerns that generally focus on the idea that the hyperparameter search ranges we use are artificially chosen instead of being general to a wide range of problems. At a high level we plan to engage with this feedback quantitatively in the next revision by including runs of the linear scaling rule over much larger search spaces to validate the robustness and benefits over other hyperparameter search methods.
> > > > >
> > > > > > the random guessing
> > > > >
> > > > > As we noted, random search works well, yet the relative error rate reduction of the linear scaling rule is still significant (in our estimation).
> > > > >
> > > > > > The $\eta$ grid feels hand-picked.
> > > > >
> > > > > We use the same grid $\eta \in [0.01, 1.00]$ for all CV datasets and only apply the original linear scaling rule as described in the preamble of `linear_scaling.py`. Based on the model-and-dataset-dependent learning rate selection done in our method, different models on different datasets may find different optimal values of $\eta$; for example this results in $\eta=20$ for ImageNet. To ensure that our results seem valid and not hand-picked, we will make sure to provide results with a larger $\eta$ grid for the camera ready.
> > > > >
> > > > > > Wouldn't you in practice have a logarithmic grid over several order of magnitude?
> > > > >
> > > > > This is a good ablation that we will do as part of the planned evaluation. One reason why we don't currently consider several orders of magnitude for each value of $\varepsilon$ is because our procedure is adaptive -given the prior that we expect $r$ to increase linearly with $\varepsilon$, we don't need to try values of $\eta$ for $\varepsilon_2=0.05$ that are much smaller than the best value for $\varepsilon_1=0.01$, and similarly for $\varepsilon_f$.
> > > > >
> > > > > > your tuning cost would increase.
> > > > >
> > > > > Although we have analyzed our choice of learning rate grid for the CV tasks in the above responses, it is possible that the cost we need to dedicate to tuning might increase for other tasks. For example, instead of $\varepsilon_1 = 0.01, \varepsilon_2 = 0.05, \varepsilon_f = 0.97$ we might need $\varepsilon_1 = 0.05, \varepsilon_2 = 0.1, \varepsilon_f = 0.96$ or even $\varepsilon_1 = 0.05, \varepsilon_2 = 0.2, \varepsilon_f = 0.9$. Or we might need to evaluate more than 2 points and fit a polynomial. Currently we err on the side of simplicity in providing our algorithm, and opt to provide a very simple heuristic algorithm ("jointly scale the learning rate and number of iterations linearly with $\varepsilon$ and fix the rest of the hyperparameters according to these rules) that works well for the tasks we consider. Future work can analyze what the granularity should be or how to allocate the privacy budget.
> > > > >
> > > > > > related to the review of BCCV
> > > > >
> > > > > We responded to your comment on the rebuttal of BCCV in the other thread; to keep things ordered we don't reproduce that response here. As a summary, we clarify that the comparison to [3] is already apples to apples; our reported results for [3] uses full batch DP-GD because we reimplement their paper in our setting to ensure an apples-to-apples comparison. Providing an apples to apples comparison to [4] is more challenging but this is a task we can undertake for the camera ready.
> > > > >
> > > > > > if you would take the method of [3] or [4] with this learning rate grid for Cifar10 or Cifar100, I believe you could get similar results.
> > > > >
> > > > > Note that the referenced [3] does not use a learning rate grid.
> > > > >
> > > > > > Looking at Appendix B
> > > > >
> > > > > In Appendix B, we provide details for the results in Section 3.3 of our main paper where we do full fine-tuning for language models. Here it is not computationally feasible to do full-batch training because we are fine-tuning the full transformers. We clarify that we can apply the linear scaling rule without using the full-batch accounting, in settings where subsampling may be necessary, by just doing the composition of heterogeneous mechanisms via the prv_accountant library. In the camera ready we will be more careful about specifying these key differences between the CV and NLP tasks.

---

### Decision · Program_Chairs · 2023-09-21

**Decision:**

Reject

**Comment:**

The reviewers are in agreement that the core techniques and ideas in this paper are interesting. However, they also raise a number of significant concerns that I think need to be addressed before the paper is published. Reviewers tXcJ and kJWb found several presentation issues that should be addressed. Reviewer kJWb also points out that a library used by the authors' code depends on the work of Bu et al. (NeurIPS 2022). This (uncited) prior work reports performance much closer to the method from this submission than any of the included baselines, making it unclear how much of the improvement comes from the work of Bu et al., and how much comes from the proposed hyperparameter optimization methodology. It seems to me that including a non-private tuning of the hyperparameters of the algorithm of Bu et al. would be a very strong baseline, along the lines of what reviewer KKeT was asking for.  Finally, reviewers BBCV and kJWb would have liked further explanation (theoretical or intuitive) for why the proposed method should work well. I believe these concerns put the paper below the bar for NeurIPS.